# GAUSSIAN-BERNOULLI RBMS WITHOUT TEARS

## ABSTRACT

We revisit the challenging problem of training Gaussian-Bernoulli restricted Boltzmann machines (GRBMs), introducing two innovations. We propose a novel Gibbs-Langevin sampling algorithm that outperforms existing methods like Gibbs sampling. We propose a modified contrastive divergence (CD) algorithm so that one can generate images with GRBMs starting from noise. This enables direct comparison of GRBMs with deep generative models, improving evaluation protocols in the RBM literature. Moreover, we show that modified CD and gradient clipping are enough to robustly train GRBMs with large learning rates, thus removing the necessity of various tricks in the literature. Experiments on Gaussian Mixtures, MNIST, FashionMNIST, and CelebA show GRBMs can generate good samples, despite their single-hidden-layer architecture.

## 1 INTRODUCTION

Restricted Boltzmann machines (RBMs) (Smolensky, 1986; Freund & Haussler, 1991; Hinton, 2002) are generative energy-based models (EBMs) with stochastic binary units. A variant of Boltzmann machines (Ackley et al., 1985), they have a bipartite graphical structure that enables efficient probabilistic inference, and they can be stacked to form deep belief networks (DBNs) (Hinton & Salakhutdinov, 2006; Bengio et al., 2006; Hinton et al., 2006) and deep Boltzmann machines (DBMs) (Salakhutdinov & Hinton, 2009; Cho et al., 2013). Gaussian-Bernoulli RBMs (GRBMs) (Welling et al., 2004; Hinton & Salakhutdinov, 2006) extend RBMs to model continuous data by replacing the binary visible units of the RBM with Gaussian random variables.

GRBMs remain challenging to learn, however, despite many proposed modifications to the model or training algorithm. For instance, Lee et al. (2007) add a regularization term to encourage sparsely activated binary hidden units. Krizhevsky et al. (2009) attribute the difficulties in learning to high-frequency noise present in natural images. Factorized high-order terms were introduced in (Ranzato & Hinton, 2010; Ranzato et al., 2010) to allow GRBMs to explicitly learn the covariance structure among pixels. Nair & Hinton (2010) suggest that binary hidden units are problematic, and proposed model variants with real-valued hidden units. Cho et al. (2011a; 2013) advocate the use of parallel tempering sampling (Earl & Deem, 2005), adaptive learning rate, and enhanced gradient (Cho et al., 2011b) to improve GRBM learning. Melchior et al. (2017) conclude that difficulties in GRBM training are due to training algorithms rather than the model itself; they advocate the use of gradient clipping, specialized weight initialization, and contrastive divergence (CD) (Hinton, 2002) rather than persistent CD (Tieleman, 2008). Tramel et al. (2018) propose the truncated Gaussian visible units and employ the Thouless-Anderson-Palmer (TAP) mean-field approximation for inference and learning. Upadhya & Sastry (2021) propose a stochastic difference of convex functions programming (S-DCP) algorithm to replace CD in training GRBMs.

An important motivation for improving GRBM learning is so that it can be used as a front-end to convert real-valued data to stochastic binary data. This would enable research on modelling real-valued data via DBMs/DBNs, which are more expressive due to their deep architectures. This class of models are of special interest: their learning algorithm involves only local computation, and thus they are more biologically plausible than EBMs trained using backprop. As GRBMs are perhaps the simplest hybrid (including both continuous and discrete random variables) EBMs, investigating the inference and learning algorithms of GRBMs would lay the foundation and inspire more future research on deep hybrid EBMs, which are useful for many applications like generating (continuous) images and their (discrete) scene graphs. Finally, RBMs and GRBMs are actively studied in quantum computing and physics (Melko et al., 2019; Ajagekar & You, 2020) since they naturally fit the

problem formulation (*e.g*., Ising models) required by many quantum computing devices. Progress on RBMs/GRBMs could potentially benefit such interdisciplinary research.

To this end, we propose improved GRBM learning methods for image data. First, we propose a hybrid Gibbs-Langevin sampling algorithm that outperforms predominant use of Gibbs sampling. To the best of our knowledge this is the first use of Langevin sampling for GRBM training (with or without Metropolis adjustment). Second, we propose a modified CD algorithm so that one can generate images with learned GRBMs starting from Gaussian noise. This enables a fair and direct comparison of GRBMs with deep generative models, something beyond the reach of existing GRBM learning methods. Third, We show that the modified CD with gradient clipping is sufficient to train GRBMs, thus removing the need for heuristics that have been crucial for existing approaches. At last, we empirically show that GRBMs can generate good samples on Gaussian Mixtures, MNIST, FashionMNIST, and CelebA, despite they have a single hidden layer.

## 2 RELATED WORK

**Learning the variances** There are two variances to be estimated in GRBM modelling. One is the intrinsic variance of the data, *e.g*., the variance of image intensities, which is fixed once the data is observed. The other is the (extrinsic) variance parameter in GRBMs, which governs the level of additional Gaussian noises added to visible units. Learning the extrinsic variance is thus necessary for generating sharp and realistic images. But small variance parameters tend to cause the energy function and its gradient to have large values, thus making the stochastic gradient estimates returned by CD numerically unstable. Most existing methods fix the variance (*e.g*., to one) to avoid this issue. Krizhevsky et al. (2009); Cho et al. (2011a) consider learning the variance using a smaller learning rate than for other parameters, obtaining much better reconstruction, thus supporting the importance of learning variances. However, many of the learned filters are still noisy and point-like. Melchior et al. (2017) learn a shared variance across all visible units, yielding improved performance, especially with large numbers of hidden units. In this work, we learn one variance parameter per visible unit and achieve much lower learned variances than existing methods, *e.g*., approximately $1e^{-5}$ on MNIST.

**Stochastic gradient estimation and learning rate** Due to the intractable log partition function of GRBMs, one often estimates the gradients of the log likelihood w.r.t. parameters via Monte Carlo. Gibbs sampling is predominant in CD learning due to its simplicity, but it mixes slowly in practice. One can refer to (Decelle et al., 2021) for a detailed study on the mixing time of CD for RBMs. This yields noisy gradient estimates which often cause training instabilities and prohibits using large learning rates. Cho et al. (2011a) explore parallel tempering with adaptive learning rates to obtain better reconstruction. Cho et al. (2013) propose enhanced gradients that are invariant to bit-flipping in hidden units. Melchior et al. (2017) show that gradient clipping and special weight initialization support robust CD learning with large learning rates. We advocate Langevin MC to improve gradients, and validate that gradient clipping does enable training with large learning rates.

**Model capacity** Theis et al. (2011) show that GRBMs are outperformed even by simple mixture models in estimating likelihoods for image data. Wang et al. (2012); Melchior et al. (2017) demonstrate that GRBMs can be expressed as either a product of experts or a constrained Gaussian mixture in the visible domain, hinting that GRBMs need more hidden units than the true number of components to fit additive mixture densities well. Krause et al. (2013); Gu et al. (2022) provide theoretical guarantees on GRBMs for universal approximation of mixtures and smooth densities. Although this shows that GRBMs are expressive, they do not lead directly to practical GRBM learning algorithms.

**Model Evaluation** Like many deep generative models, evaluating GRBMs is difficult, as the log likelihood is intractable. To date, GRBMs have been evaluated by visually inspecting reconstructed images, filters and hidden activation (*i.e*., features), and sampled images during CD training. Quantitative metrics include reconstruction errors, and error rates of post-hoc trained classifiers on learned features. However, these metrics do not necessarily indicate if GRBMs are good generative models (Melchior et al., 2017). Unlike existing work, we sample from learned GRBMs, starting from Gaussian noise, enabling direct comparisons with other generative models, qualitatively (visually inspecting samples) and quantitatively (*e.g*., Frechet Inception distance (FID) (Heusel et al., 2017). Note that similar noise-initialization strategy has been studied in EBMs (Nijkamp et al., 2019).

---

**Algorithm 1** Langevin Sampling for GRBMs

---

1: **Input**: $\mathbf{v}^{(0)}$, step size $\alpha_0$, total step $T$, burn-in step $\tilde{T}$, adjust step $\eta$
2: **For** $t = 1, \ldots, T$
3: $\qquad \alpha_t = \text{CosineScheduler}(t, T, \alpha_0)$
4: $\qquad \mathbf{v} = \mathbf{v}^{(t-1)} - \alpha_t \frac{\partial \tilde{E}(\mathbf{v}^{(t-1)})}{\partial \mathbf{v}} + \sqrt{2\alpha_t}\boldsymbol{\xi}_t , \quad \boldsymbol{\xi}_t \sim \mathcal{N}(0, I)$ $\quad \triangleright$ Use marginal energy in Eq. (5)
5: $\qquad$ **If** $t <= \eta$ or $\left(t > \eta \text{ and } u \sim \mathcal{U}(0, 1) < A(\mathbf{v}, \mathbf{v}^{(t-1)})\right)$
6: $\qquad\qquad \mathbf{v}^{(t)} = \mathbf{v}$
7: $\qquad$ **Else**
8: $\qquad\qquad \mathbf{v}^{(t)} = \mathbf{v}^{(t-1)}$
9: **Return**: $\{\mathbf{v}^{(\tilde{T}+1:T)}\}$ $\qquad\qquad\qquad \triangleright i : j$ indexes consecutive samples from $i$-th to $j$-th

---

## 3 GAUSSIAN-BERNOULLI RESTRICTED BOLTZMANN MACHINES

A Gaussian-Bernoulli Restricted Boltzmann Machine (GRBM) (Welling et al., 2004; Krizhevsky et al., 2009; Cho et al., 2011a; Melchior et al., 2017) is a Markov Random Field (MRF) with continuous stochastic visible units and binary stochastic hidden units. Denoting $N$ visible units as $\mathbf{v} \in \mathbb{R}^N$ and $M$ hidden units as $\mathbf{h} \in \{0, 1\}^M$, the energy function associated with a GRBM is defined to be

$$E_\theta(\mathbf{v}, \mathbf{h}) = \frac{1}{2} \left(\frac{\mathbf{v} - \boldsymbol{\mu}}{\boldsymbol{\sigma}}\right)^\top \left(\frac{\mathbf{v} - \boldsymbol{\mu}}{\boldsymbol{\sigma}}\right) - \left(\frac{\mathbf{v}}{\boldsymbol{\sigma}^2}\right)^\top W\mathbf{h} - \mathbf{b}^\top \mathbf{h} , \tag{1}$$

with weight matrix $W \in \mathbb{R}^{N \times M}$, bias $\boldsymbol{b} \in \mathbb{R}^M$, mean $\boldsymbol{\mu} \in \mathbb{R}^N$, and variance $\boldsymbol{\sigma}^2 \in \mathbb{R}^N_+$, where, unless stated otherwise, $\frac{\mathbf{x}}{\mathbf{y}}$ denotes element-wise division between vectors $\mathbf{x}$ and $\mathbf{y}$, as is convention in the GRBM literature. We denote the set of learnable parameters as $\theta = \{W, \boldsymbol{b}, \boldsymbol{\mu}, \boldsymbol{\sigma}^2\}$. To ensure the variance remains non-negative during learning, we adopt a reparameterization, directly learning $\log \boldsymbol{\sigma}^2$ rather than $\boldsymbol{\sigma}^2$ or $\boldsymbol{\sigma}$. Finally, given the energy function, one can define the Boltzmann distribution, over visible and hidden states, as

$$p_\theta(\mathbf{v}, \mathbf{h}) = \frac{1}{Z} \exp\left(-E_\theta(\mathbf{v}, \mathbf{h})\right) , \quad \text{where} \quad Z = \int_{-\infty}^{+\infty} \sum_{\mathbf{h}} \exp\left(-E_\theta(\mathbf{v}, \mathbf{h})\right) \mathrm{d}\mathbf{v} \tag{2}$$

is the normalization constant, which is intractable for even moderately large $M$.

The underlying graphical model, like an RBM, is a bipartite graph with edges only connecting visible units to hidden units. This entails conditional independence of the form $p(\mathbf{v}|\mathbf{h}) = \prod_i p(\mathbf{v}_i|\mathbf{h})$ and $p(\mathbf{h}|\mathbf{v}) = \prod_j p(\mathbf{h}_j|\mathbf{v})$. One can also derive the following conditional distributions for GRBMs,

$$p(\mathbf{v}|\mathbf{h}) = \mathcal{N}\left(\mathbf{v}|W\mathbf{h} + \boldsymbol{\mu}, \text{diag}(\boldsymbol{\sigma}^2)\right) \quad (3) \qquad p(\mathbf{h}_j = 1|\mathbf{v}) = \left[\text{Sigmoid}\left(W^\top \frac{\mathbf{v}}{\boldsymbol{\sigma}^2} + \mathbf{b}\right)\right]_j , \quad (4)$$

where $\mathcal{N}\left(\mathbf{v}|W\mathbf{h} + \boldsymbol{\mu}, \text{diag}(\boldsymbol{\sigma}^2)\right)$ is the multivariate Gaussian distribution with mean $W\mathbf{h} + \boldsymbol{\mu}$, and the diagonal covariance matrix $\text{diag}(\boldsymbol{\sigma}^2)$. Here, $\text{Sigmoid}(\mathbf{x}) = 1/(1 + \exp(-\mathbf{x}))$ is applied to the vector $\mathbf{x}$ in an element-wise manner, and $[\cdot]_j$ denotes the $j$-th element of the corresponding vector.

The marginal distribution over visible units is $p(\mathbf{v}) = \exp(-\tilde{E}_\theta(\mathbf{v}))/Z$, where

$$\tilde{E}_\theta(\mathbf{v}) = \frac{1}{2} \left(\frac{\mathbf{v} - \boldsymbol{\mu}}{\boldsymbol{\sigma}}\right)^\top \left(\frac{\mathbf{v} - \boldsymbol{\mu}}{\boldsymbol{\sigma}}\right) - \text{Softplus}\left(W^\top \frac{\mathbf{v}}{\boldsymbol{\sigma}^2} + \mathbf{b}\right)^\top \mathbf{1} , \tag{5}$$

and $\text{Softplus}(\mathbf{x}) = \log(1 + \exp(\mathbf{x}))$ is applied in an element-wise manner, and $\mathbf{1}$ is the all-one vector of size $M$. We call $\tilde{E}_\theta(\mathbf{v})$ the *marginal energy* to distinguish it from the GRBM energy in Eq. (1). We leave the derivation to Appendix A.1. As shown in Melchior et al. (2017), one can also rewrite the marginal distribution $p(\mathbf{v})$ as a constrained Gaussian mixture.

### 3.1 INFERENCE

When performing probabilistic inference, one often chooses between variational inference (Hinton & Van Camp, 1993; Jordan et al., 1999) and Markov chain Monte Carlo (MCMC) methods (Neal, 1993; Andrieu et al., 2003). We focus on MCMC as common variational methods have been less

effective with RBMs and GRBMs (Gabrié et al., 2015; Takahashi & Yasuda, 2016). From the generative modelling perspective, we wish to draw samples of visible units during inference. There are two natural approaches to this: 1) sample from the joint distribution in Eq. (2) and discard the samples of hidden units, or 2) directly sample from the marginal distribution.

Gibbs sampling (Geman & Geman, 1984) is perhaps the predominant approach, due to its simplicity. In the context of GRBMs, one alternates between sampling hidden units given visible units, and sampling visible units given hidden units. This produces samples from the joint distribution in Eq. (2). The detailed Gibbs sampling algorithm is given in Appendix A.2.

**Langevin Sampling.** Langevin Monte Carlo (Grenander & Miller, 1994; Roberts & Tweedie, 1996; Welling & Teh, 2011) is a class of MCMC methods that generate samples from a distribution of continuous random variables by simulating Langevin dynamics. Since GRBMs are hybrid EBMs, *i.e.*, comprising continuous and discrete random variables, we have at least two ways to leverage Langevin sampling. One is to directly apply Langevin sampling to the marginal distribution of visible units in Eq. (5). Suppose at time step $t-1$, we have sample $\mathbf{v}_{t-1}$ and want to draw a new sample $\mathbf{v}_t$. The proposal distribution corresponding to one-step Langevin dynamics is given by

$$q(\mathbf{v}|\mathbf{v}^{(t-1)}) = \mathcal{N}\left(\mathbf{v}\left|\mathbf{v}^{(t-1)} - \alpha_t \frac{\partial \tilde{E}(\mathbf{v}^{(t-1)})}{\partial \mathbf{v}}, 2\alpha_t I\right.\right), \tag{6}$$

where the gradient the of marginal energy $\tilde{E}$ w.r.t. the visible units is given in Appendix A.3. If we use the Metropolis-Hastings algorithm to accept or reject proposed samples, the acceptance probability of a proposal $\mathbf{v}_t$, given the previous state, $\mathbf{v}_{t-1}$, is (see Appendix A.3 for derivation):

$$A(\mathbf{v}^{(t)}, \mathbf{v}^{(t-1)}) = \min\left(1, \frac{\exp\left(-\tilde{E}_\theta(\mathbf{v}^{(t)}) - \frac{1}{4\alpha_t}\left\|\mathbf{v}^{(t-1)} - \mathbf{v}^{(t)} + \alpha_t \frac{\partial \tilde{E}(\mathbf{v}^{(t)})}{\partial \mathbf{v}}\right\|^2\right)}{\exp\left(-\tilde{E}_\theta(\mathbf{v}^{(t-1)}) - \frac{1}{4\alpha_t}\left\|\mathbf{v}^{(t)} - \mathbf{v}^{(t-1)} + \alpha_t \frac{\partial \tilde{E}(\mathbf{v}^{(t-1)})}{\partial \mathbf{v}}\right\|^2\right)}\right). \tag{7}$$

Alg. 1 shows the Metropolis-adjusted Langevin Algorithm (MALA) for the marginal GRBM. Compared to generic MALA, it also includes an extra hyperparameter, namely, the adjust step $\eta$. If $\eta$ is set to 0, then we perform a Metropolis adjustment at every sampling step, as prescribed in the generic MALA. If $\eta$ is set to $K > 0$, then we skip the Metropolis adjustment for the first $K$ steps. The adjust step effectively controls a trade-off between sampling accuracy[1] and computational efficiency. Since we do not hope to see Gaussian noise in our final-sampled images (*i.e.*, beyond the level of intrinsic noise in the observations), it is beneficial to decay the noise level, as in score-based models (Song & Ermon, 2019). For certain step-size-annealing schedules and energy functions, there are theoretical guarantees on the convergence of Langevin sampling (Durmus & Moulines, 2019). For simplicity, we use the cosine scheduler and find it works well in practice. More details about the scheduler are provided in Appendix A.3.

**Gibbs-Langevin Sampling.** We also introduce a new hybrid sampler for GRBMs (see Alg. 2). Like the Gibbs sampler, it alternates between sampling hidden units conditioned on visible units, and sampling visible units given the hidden units. Unlike generic Gibbs, which directly samples from the Gaussian $p(\mathbf{v}|\mathbf{h}^{(t)})$, we instead use Langevin MC to sample the continuous visible units given the hidden state. The use of Langevin MC may seem unnecessary because the Gaussian conditional permits a one-step sampling algorithm. The subtlety comes from the fact that the finite-step Langevin sampler explicitly depends on the initial sample. Specifically, the proposal distribution of one complete outer-loop step in Alg. 2, *e.g.*, at iteration $t-1$, can be expressed as

$$q(\mathbf{v}, \mathbf{h}|\mathbf{v}^{(t-1)}, \mathbf{h}^{(t-1)}) = q(\mathbf{h}|\mathbf{v}) q(\mathbf{v}|\mathbf{v}^{(t-1)}, \mathbf{h}^{(t-1)}), \tag{8}$$

where $q(\mathbf{h}|\mathbf{v})$ is given by Eq. (4), and $q(\mathbf{v}|\mathbf{v}^{(t-1)}, \mathbf{h}^{(t-1)})$ is the proposal distribution of a K-step Langevin sampler (*i.e.* from the inner loop). This proposal distribution explicitly depends on the initial visible sample, $\mathbf{v}^{(t-1)}$ from iteration $t-1$. By contrast, the generic Gibbs sampler does not have such dependence, *i.e.*, $q(\mathbf{v}|\mathbf{v}^{(t-1)}, \mathbf{h}^{(t-1)}) = q(\mathbf{v}|\mathbf{h}^{(t-1)})$. This dependence allows us to construct a persistent Markov chain in the space of visible units. Moreover, the Langevin sampler

---

[1]Here sampling accuracy means the closeness between the underlying distribution of samples and the target distribution measured in, *e.g.*, total variation or Wasserstein distances.

---

**Algorithm 2** Gibbs-Langevin Sampling for GRBMs

---

1: **Input**: $\mathbf{v}^{(0)}, \mathbf{h}^{(0)}$, step size $\alpha_0$, total step $T$, burn-in step $\tilde{T}$, adjust step $\eta$, Langevin step $K$
2: **Function** Langevin($\tilde{\mathbf{v}}^{(0)}, \mathbf{h}, \alpha_0, \mathbf{K}$):
3:      **For** $k = 1, \ldots, K$
4:          $\alpha_k = \text{CosineScheduler}(k, K, \alpha_0)$
5:          $\tilde{\mathbf{v}}^{(k)} = \tilde{\mathbf{v}}^{(k-1)} - \alpha_k \frac{\partial E(\tilde{\mathbf{v}}^{(k-1)}, \mathbf{h})}{\partial \mathbf{v}} + \sqrt{2\alpha_k}\boldsymbol{\xi}_k$,    $\boldsymbol{\xi}_k \sim \mathcal{N}(0, I)$
6: **Return** $\tilde{\mathbf{v}}^{(K)}$
7:
8: **For** $t = 1, \ldots, T$
9:      $\mathbf{v} = \text{Langevin}(\mathbf{v}^{(t-1)}, \mathbf{h}^{(t-1)}, \alpha_0, K)$
10:      $\mathbf{h} \sim p(\mathbf{h}|\mathbf{v})$
11:      **If** $t <= \eta$ or $\left( t > \eta \text{ and } u \sim \mathcal{U}(0,1) < \tilde{A}\left((\mathbf{v}, \mathbf{h}), (\mathbf{v}^{(t-1)}, \mathbf{h}^{(t-1)})\right) \right)$
12:          $\mathbf{v}^{(t)}, \mathbf{h}^{(t)} = \mathbf{v}, \mathbf{h}$
13:      **Else**
14:          $\mathbf{v}^{(t)}, \mathbf{h}^{(t)} = \mathbf{v}^{(t-1)}, \mathbf{h}^{(t-1)}$
15: **Return**: $\{(\mathbf{v}^{(\tilde{T}+1:T)}, \mathbf{h}^{(\tilde{T}+1:T)})\}$

---

leverages the informative gradient of log density whereas Gibbs sampler does not. We find that our new sampler performs significantly better than the vanilla Gibbs sampler in practice.

The Metropolis adjustment for these Gibbs-Langevin proposals is, however, somewhat more involved. Following Alg. 2, with $\tilde{\mathbf{v}}^{(0)} = \mathbf{v}^{(t-1)}$ and $\tilde{\mathbf{v}}^{(K)} = \mathbf{v}$, by marginalizing out the intermediate states on the Markov chain, we obtain the proposal

$$q(\tilde{\mathbf{v}}^{(K)}|\tilde{\mathbf{v}}^{(0)}, \mathbf{h}^{(t-1)}) = \int \cdots \int \left( \prod_{k=1}^{K} q(\tilde{\mathbf{v}}^{(k)}|\tilde{\mathbf{v}}^{(k-1)}, \mathbf{h}^{(t-1)}) \right) \mathrm{d}\tilde{\mathbf{v}}^{(1)} \cdots \mathrm{d}\tilde{\mathbf{v}}^{(K-1)}. \quad (9)$$

The integrand in Eq. 9 comprises $K$ one-step Langevin updates, each of which is given by

$$q(\mathbf{v}|\tilde{\mathbf{v}}^{(k-1)}, \mathbf{h}^{(t-1)}) = \mathcal{N}\left( \mathbf{v}\bigg|\tilde{\mathbf{v}}^{(k-1)} - \alpha_k \frac{\partial E(\tilde{\mathbf{v}}^{(k-1)}, \mathbf{h}^{(t-1)})}{\partial \mathbf{v}}, 2\alpha_k I \right), \quad (10)$$

for which the energy gradient is given in Appendix A.4. Although the multiple integral in Eq. (9) appears intractable, one can use reparameterization to derive the following analytical form,

$$q(\tilde{\mathbf{v}}^{(K)}|\tilde{\mathbf{v}}^{(0)}, \mathbf{h}^{(t-1)}) = \mathcal{N}\left( \boldsymbol{\beta}_0\tilde{\mathbf{v}}^{(0)} + \left( \sum_{k=1}^{K} \boldsymbol{\beta}_k\alpha_k \right) \frac{\boldsymbol{\mu} + W\mathbf{h}^{(t-1)}}{\boldsymbol{\sigma}^2}, \text{diag}\left( \sum_{k=1}^{K} 2\alpha_k\boldsymbol{\beta}_k^2 \right) \right), \quad (11)$$

where $\boldsymbol{\beta}_k = \prod_{j=k+1}^{K}\left(1 - \frac{\alpha_j}{\boldsymbol{\sigma}^2}\right), \forall k \in \{0, \ldots, K-1\}$ and $\boldsymbol{\beta}_K = \mathbf{1}$. Based on this result, one can show that the acceptance probability for the Metropolis adjustment is

$$\tilde{A}((\mathbf{v}^{(t)}, \mathbf{h}^{(t)}), (\mathbf{v}^{(t-1)}, \mathbf{h}^{(t-1)})) =$$

$$\min\left( 1, \frac{\exp\left( -E_\theta(\mathbf{v}^{(t)}, \mathbf{h}^{(t)}) - \left\| \frac{\mathbf{v}^{(t-1)} - \boldsymbol{\beta}_0\mathbf{v}^{(t)} - \boldsymbol{a}(\boldsymbol{\mu} + W\mathbf{h}^{(t)})}{\sqrt{2}\tilde{\boldsymbol{\sigma}}} \right\|^2 \right) q(\mathbf{h}^{(t-1)}|\mathbf{v}^{(t-1)})}{\exp\left( -E_\theta(\mathbf{v}^{(t-1)}, \mathbf{h}^{(t-1)}) - \left\| \frac{\mathbf{v}^{(t)} - \boldsymbol{\beta}_0\mathbf{v}^{(t-1)} - \boldsymbol{a}(\boldsymbol{\mu} + W\mathbf{h}^{(t-1)})}{\sqrt{2}\tilde{\boldsymbol{\sigma}}} \right\|^2 \right) q(\mathbf{h}^{(t)}|\mathbf{v}^{(t)})} \right), \quad (12)$$

where $q(\mathbf{h}_j = 1|\mathbf{v}) = \left[ \text{Sigmoid}\left( W^\top \frac{\mathbf{v}}{\boldsymbol{\sigma}^2} + \mathbf{b} \right) \right]_j$, $\boldsymbol{a} = \frac{\sum_{k=1}^{K} \boldsymbol{\beta}_k\alpha_k}{\boldsymbol{\sigma}^2}$, and $\tilde{\boldsymbol{\sigma}}^2 = \sum_{k=1}^{K} 2\alpha_k\boldsymbol{\beta}_k^2$. We leave derivations to Appendix A.4.

### 3.2 LEARNING

To learn GRBMs, we maximize the log likelihood of the observed data using stochastic gradient-based methods, *e.g.*, contrastive divergence (CD). Depending on whether we use the joint (Eq. (2)) or the marginal (Eq. (5)) distribution, we have two possible gradient estimators.

**Learning with the Joint Distribution.** When optimizing the GRBM with the joint distribution, one can express the general form of the gradient of the log likelihood w.r.t. parameters $\theta$ as

$$\nabla\theta = \left\langle -\frac{\partial E_\theta(\mathbf{v}, \mathbf{h})}{\partial \theta} \right\rangle_d - \left\langle -\frac{\partial E_\theta(\mathbf{v}, \mathbf{h})}{\partial \theta} \right\rangle_m. \quad (13)$$

---

**Algorithm 3** Modified CD Learning Algorithm for GRBMs with Joint Density

---

1: **Input**: CD-step $K$, burn-in step $M$, learning Rate $\eta$, Langevin step size $\alpha_0$, SGD step $T$
2: **For** $t = 1, \cdots, T$
3:      $\mathbf{v}^+ = \mathbf{v}_{\text{data}}$
4:      $\mathbf{h}^+ \sim p(\mathbf{h}|\mathbf{v}^+)$
5:      $\nabla\theta^+ = \left\langle \frac{\partial E(\mathbf{v}^+, \mathbf{h}^+)}{\partial\theta} \right\rangle_d$               ▷ Compute Positive Gradient
6:      $\mathbf{v}_0^- \sim \mathcal{N}(\mathbf{0}, I), \mathbf{h}_0^- \sim p(\mathbf{h}|\mathbf{v}_0^-)$         ▷ Modified CD to start with noise
7:      $\{\mathbf{v}_{M:K}^-, \mathbf{h}_{M:K}^-\} \sim \text{Sampler}(\mathbf{v}_0^-, \mathbf{h}_0^-, \alpha_0\bar{\boldsymbol{\sigma}})$    ▷ Alg. 5 or Alg. 2, $\bar{\boldsymbol{\sigma}}$ is current mean variance
8:      $\nabla\theta^- = \left\langle \frac{\partial E(\mathbf{v}^-, \mathbf{h}^-)}{\partial\theta} \right\rangle_m$               ▷ Compute Negative Gradient
9:      $\theta = \theta - \eta(\nabla\theta^+ - \nabla\theta^-)$                   ▷ Compute Update
10: **Return** $\theta$

---

Here, following the notation in the RBM literature, we denote expectation under the *data distribution*, *i.e.*, $p_\theta(\mathbf{h}|\mathbf{v}) p_{\text{data}}(\mathbf{v})$, as $\langle\cdot\rangle_d = \mathbb{E}_{p_\theta(\mathbf{h}|\mathbf{v})p_{\text{data}}(\mathbf{v})}[\cdot]$. Similarly, we denote the expectation under the *model distribution*, $p_\theta(\mathbf{v}, \mathbf{h})$ as $\langle\cdot\rangle_m = \mathbb{E}_{p_\theta(\mathbf{v}, \mathbf{h})}[\cdot]$. The expected gradients under the data and model distributions are called *positive* and *negative* gradients respectively. Based on Eq. (13), we can formulate the gradients of specific parameters as follows,

$$\nabla W_{ij} = \left\langle \frac{\mathbf{v}_i}{\boldsymbol{\sigma}_i^2}\mathbf{h}_j \right\rangle_d - \left\langle \frac{\mathbf{v}_i}{\boldsymbol{\sigma}_i^2}\mathbf{h}_j \right\rangle_m \tag{14}$$

$$\nabla\mu_i = \left\langle \frac{\mathbf{v}_i - \mu_i}{\boldsymbol{\sigma}_i^2} \right\rangle_d - \left\langle \frac{\mathbf{v}_i - \mu_i}{\boldsymbol{\sigma}_i^2} \right\rangle_m \tag{15}$$

$$\nabla\log\boldsymbol{\sigma}_i^2 = \left\langle \frac{(\mathbf{v}_i - \mu_i)^2}{2\boldsymbol{\sigma}_i^2} - \frac{\sum_j \mathbf{v}_i W_{ij}\mathbf{h}_j}{\boldsymbol{\sigma}_i^2} \right\rangle_d - \left\langle \frac{(\mathbf{v}_i - \mu_i)^2}{2\boldsymbol{\sigma}_i^2} - \frac{\sum_j \mathbf{v}_i W_{ij}\mathbf{h}_j}{\boldsymbol{\sigma}_i^2} \right\rangle_m \tag{16}$$

$$\nabla\mathbf{b}_i = \langle\mathbf{h}_i\rangle_d - \langle\mathbf{h}_i\rangle_m . \tag{17}$$

Since the expectations in these gradients are generally intractable, we use Monte Carlo methods to approximate them. To sample from the joint density, we can use Gibbs or Gibbs-Langevin samplers as described in Sec. 3.1. The overall learning algorithm is outlined in Alg. 3. An important detail is that we multiply the initial Langevin step size by the average variance at each gradient update step and then feed it to the sampler. Since the variance is decreasing (the energy function and its gradient are increasing) as learning goes on, keeping the step size roughly invariant to such scaling would make the sampling more effective.

**Learning with the Marginal Distribution.** Now we turn to learning the model under the marginal distribution in Eq. (5). Since we have the marginal distribution of visible units, we can directly get the gradients of log likelihood w.r.t. model parameters $\theta$ as,

$$\nabla\theta = \left\langle -\frac{\partial\tilde{E}_\theta(\mathbf{v})}{\partial\theta} \right\rangle_d - \left\langle -\frac{\partial\tilde{E}_\theta(\mathbf{v})}{\partial\theta} \right\rangle_m . \tag{18}$$

Since the gradient $\frac{\partial\tilde{E}_\theta(\mathbf{v})}{\partial\theta}$ does not depend on $\mathbf{h}$ anymore, we have

$$\left\langle -\frac{\partial\tilde{E}_\theta(\mathbf{v})}{\partial\theta} \right\rangle_d = \mathbb{E}_{p_{\text{data}}(\mathbf{v})}\left[ -\frac{\partial\tilde{E}_\theta(\mathbf{v})}{\partial\theta} \right], \qquad \left\langle -\frac{\partial\tilde{E}_\theta(\mathbf{v})}{\partial\theta} \right\rangle_m = \mathbb{E}_{p_\theta(\mathbf{v})}\left[ -\frac{\partial\tilde{E}_\theta(\mathbf{v})}{\partial\theta} \right]. \tag{19}$$

Based on above results, we can work out the detailed gradients which are the same as those in Eq. (14) to Eq. (17) but with $\mathbf{h}$ replaced with Sigmoid $\left(W^\top\frac{\mathbf{v}}{\boldsymbol{\sigma}^2} + \mathbf{b}\right)$. More details are left to Appendix A.5. We use the Langevin sampler in Sec. 3.1 to sample from the marginal density to approximate the intractable expectation. The overall learning algorithm is outlined in Alg. 4.

**Modified Contrastive Divergence.** The above two learning algorithms resemble CD if one ignores the specific sampler used. There exists a subtle yet important difference however. For most deep generative models one generates samples starting from noise. But this does not work well for models trained with CD, where sampling starts from observed data. This discrepancy of the starting sample between training and testing would be a significant issue if the Markov chain does not mix

---

**Algorithm 4** Modified CD Learning Algorithm for GRBMs with Marginal Density

---

1: **Input**: CD-step $K$, burn-in step $M$, learning rate $\eta$, Langevin step size $\alpha_0$, SGD step $T$
2: **For** $t = 1, \cdots, T$
3:      $\mathbf{v}^+ = \mathbf{v}_{\text{data}}$
4:      $\nabla\theta^+ = \left\langle \frac{\partial \tilde{E}(\mathbf{v}^+)}{\partial \theta} \right\rangle_d$                 ▷ Compute Positive Gradient
5:      $\mathbf{v}_0^- \sim \mathcal{N}(\mathbf{0}, I)$                 ▷ Modified CD to start with noise
6:      $\{\mathbf{v}_i^- | i = M, \cdots, K\} \sim \text{Sampler}(\mathbf{v}_0^-, \alpha_0\bar{\boldsymbol{\sigma}})$     ▷ Alg. 1, $\bar{\boldsymbol{\sigma}}$ is current mean variance
7:      $\nabla\theta^- = \left\langle \frac{\partial \tilde{E}(\mathbf{v}^-)}{\partial \theta} \right\rangle_m$                 ▷ Compute Negative Gradient
8:      $\theta = \theta - \eta(\nabla\theta^+ - \nabla\theta^-)$              ▷ Compute Update
9: **Return** $\theta$

---

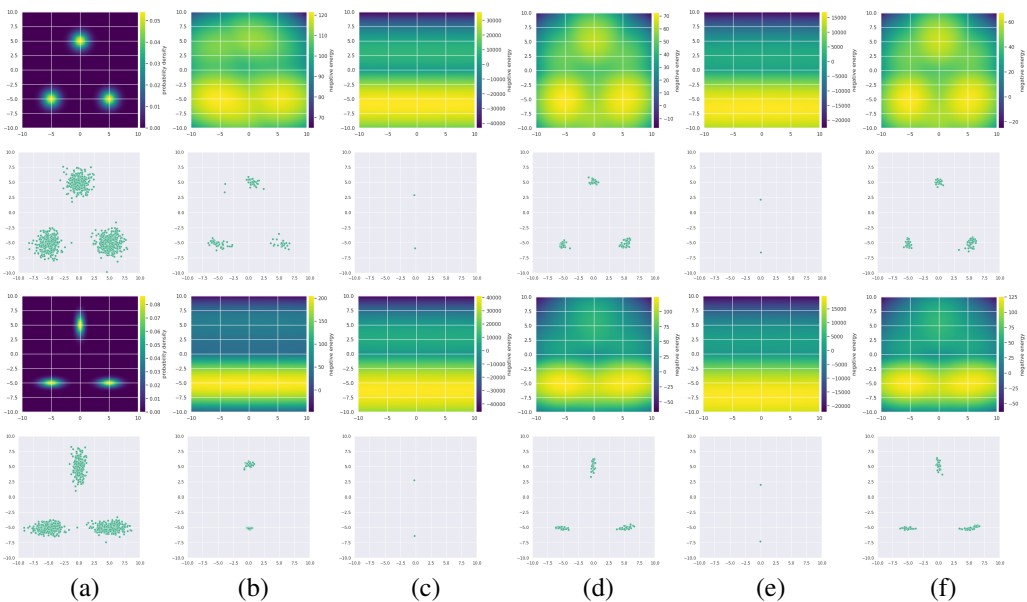

(a)           (b)           (c)           (d)           (e)           (f)

Figure 1: Density modelling using GRBMs on data from a Gaussian mixtures with isotropic (rows 1 and 2) and anisotropic variances (rows 3 and 4). Rows 1 and 3 show normalized GMM densities and (unnormalized) negative energy values for GRBMs. Rows 2 and 4 show samples drawn under different models and methods; *i.e.*, (a) Ground Truth; (b) Gibbs; (c) Langevin wo. Adjust; (d) Langevin w. Adjust; (e) Gibbs-Langevin wo. Adjust; (f) Gibbs-Langevin w. Adjust.

sufficiently quickly. We therefore modify CD by running two Markov chains to collect samples for positive and negative gradients respectively. The positive Markov chain is the same as in CD, *i.e.*, starting from observed data. The negative Markov chain now starts from a sample of standard Normal noise rather than the reconstructed data[2]. Since the positive chain starting from data will usually stay close to the data distribution, this modification pushes the negative Markov chain, starting from noise, toward the data distribution. Moreover, the discrepancy between training and testing ceases to be important as we can start from standard Normal noise while sampling from the learned model.

## 4   EXPERIMENTS

We examine the empirical behavior of our new GRBM algorithms on benchmark image datasets, namely, MNIST, Fashion-MNIST (Xiao et al., 2017), and CelebA (Liu et al., 2015).

**Implementation Details.** We found that training with modified CD alone occasionally diverges, necessitating careful tuning of the learning rate. However, adding gradient clipping (*e.g.*, clip gradient norm to 10) enables stable training with all aforementioned sampling methods. We therefore set

---

[2]The reconstructed data is typically obtained by running one complete step of Gibbs sampler from the observed data, thus being highly likely close to observed data.

learning rate to $0.01$ for all experiments. Such a large learning rate almost never works in the literature. Melchior et al. (2017) used gradient clipping and similarly large learning rates, but they had to set the learning rate for the variances 100 times smaller than that for the weights and biases during CD training. But thanks to the modified CD and gradient clipping, we found this special treatment of variances is unnecessary. We do not use momentum, weight decay, PCD, or other tricks.

### 4.1 Modeling Gaussian Mixture Densities

We first evaluate density modelling by GRBMs when the data density is known, *i.e.*, Gaussian mixture models (GMMs) in our case. This is challenging for GRBMs as the marginal distribution of visible units of GRBMs is essentially a constrained Gaussian mixture, *i.e.*, the weights of mixture components depend on one another (Melchior et al., 2017). As such, the mixture components in GRBMs can not be freely placed in the visible domain so one actually needs more hidden units than the log of the number of mixture components to fit GMMs well. We consider the 2D case for simplicity and better visibility. We generate 1,000 samples from two types (isotropic and anisotropic variances) of GMMs with 3 components as shown in Fig. 1, and learn GRBMs using our modified CD with different sampling algorithms, from which we can draw samples. Here all samplers run for 100 steps during both CD training and testing (see Appendix B.1 for more detail). Density plots and samples are shown in Fig. 1. Notice that Gibbs manages to recover the three modes in the isotropic case but fails in the anisotropic case. Both Langevin and Gibbs-Langevin sampling collapse when the adjustment is absent. We believe the cosine step size schedule contributes to the collapse as it removes more stochasticity of Langevin dynamics with small step sizes, thus making sampling more similar to gradient descent. But as we will see later, in image modelling, this may not be so severe; there are more modes so that the sampling may collapse to different modes, and the diversity of images remains acceptable. Finally, both Langevin and Gibbs-Langevin do recover all three modes with the adjustment, which shows the adjustment helps the mixing in this synthetic case.

### 4.2 Image Generation

We learn GRBMs to fit image datasets including MNIST, FashionMNIST, and CelebA. To the best of our knowledge, this is the first time that GRBMs have been shown to (unconditionally) generate good images. We provide the ablation study in Appendix B.2 and more results in Appendix B.3.

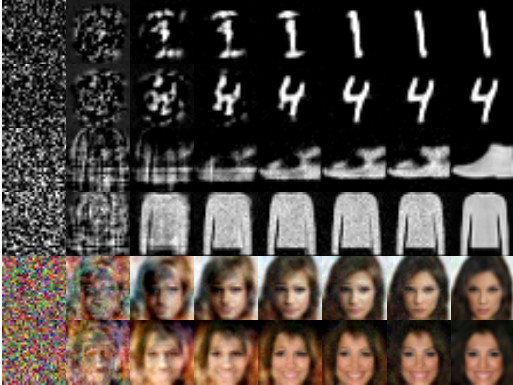

Figure 2: Intermediate samples from Gibbs-Langevin sampling.

| Methods | FID |
|---|---|
| VAE | 16.13 |
| 2sVAE (Dai & Wipf, 2019) | 12.60 |
| PixelCNN++ (Salimans et al.) | 11.38 |
| WGAN (Arjovsky et al., 2017) | 10.28 |
| NVAE (Vahdat & Kautz, 2020) | 7.93 |
| GRBMs | |
| Gibbs | 47.53 |
| Langevin wo. Adjust | 43.80 |
| Langevin w. Adjust | 41.24 |
| Gibbs-Langevin wo. Adjust | 17.49 |
| Gibbs-Langevin w. Adjust | 19.27 |

Table 1: Results on MNIST dataset.

**MNIST.** We train GRBMs with hidden size 4096 and 100 sampling steps on MNIST. We compare FID scores of GRBMs with other deep generative models in Table 1. We can see that Gibbs-Langevin family works significantly better than the Langevin family. The Metropolis adjustment ("w. Adjust" in Table 1) improves Langevin slightly but degrades Gibbs-Langevin slightly, which is different from what we observed on synthetic data. This is likely because the image distribution is so complicated (*e.g.*, having significantly more modes) that the adjustment rejects proposed moves more frequently than before. Some sophisticated strategy may be needed to increase the acceptance probability. Nevertheless, GRBMs trained with Gibbs-Langevin without adjustment achieve FID scores comparable to other generative models, which is impressive given the single-hidden-layer architecture. The learning curve of (natural) log variance is shown in Fig. 3a. The learned variance

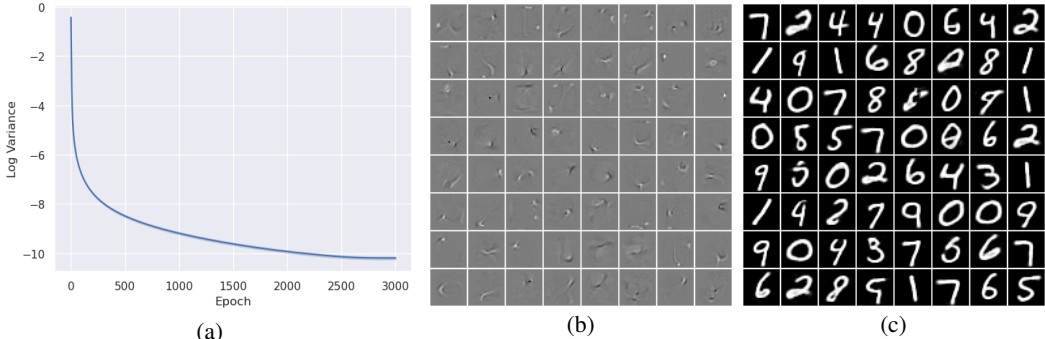

Figure 3: (a) Learning curve of (natural) log variances, (b) learned filters, and (c) samples on MNIST.

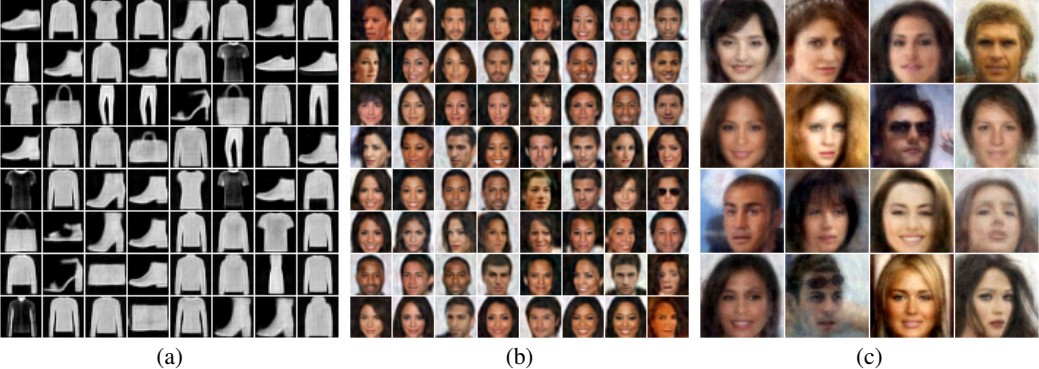

Figure 4: Samples from GRBMs on (a) FashionMNIST, (b) CelebA-32, and (c) CelebA-2K-64.

converges to around $1e^{-5}$ which is significantly smaller than those reported in the literature. The learned filters are shown in Fig. 3b. Although some point-like filters still exist, stroke-like filters are common, thus indicating GRBMs indeed learn meaningful features. We show samples drawn from the best GRBM in Fig. 3c. The intermediate samples from Gibbs-Langevin are shown in Fig. 2. Since Gibbs-Langevin without adjustment works the best, we use it for remaining experiments.

**FashionMNIST.** We then train GRBMs on FahsionMNIST which is more challenging than MNIST. We set hidden size to 10,000 and the sampling step to 100. Samples drawn from learned GRBMs are shown in Fig. 4a. GRBMs successfully learn the shapes of clothes, shoes, bags, and so on. However, they fail to capture fine textures. Since many images in this dataset look similar in shape but differ in texture, the resulting samples look similar to each other.

**CelebA.** Last, we consider the even more challenging CelebA dataset. In particular, we explore two versions of this dataset: 1) CelebA-32 where we center-crop ($140 \times 140$) and downsample images to $32 \times 32$; 2) CelebA-2K-64 where randomly select 2,000 images from the original CelebA and apply the same center crop and downsampling to $64 \times 64$. We set hidden size to 10,000 and explore the number of 100 and 200 sampling steps. Generated samples are shown in Fig. 4b and 4c. From the figure, we can see that GRBMs can learn to generate reasonably good face images.

## 5 CONCLUSION

In this paper, we revisit learning Gaussian-Bernoulli restricted Boltzmann machines. We investigate Langevin Monte Carlo and propose a novel Gibbs-Langevin sampling method. Furthermore, we modify the contrastive divergence (CD) algorithm so that one can sample data from learned GRBMs starting from noise. Modified CD along with gradient clipping enables robust training of GRBMs with large learning rates. Finally, we show that GRBMs can unconditionally generate images with good qualities, despite its single-hidden-layer architecture. In the future, it would be beneficial to extend the current GRBMs to convolutional GRBMs which should be able to learn better localized filters. Meanwhile, it would be interesting to explore Gaussian deep belief networks (GDBNs), which are deeper than GRBMs and should be superior. At last, investigating our Gibbs-Langevin sampling for hybrid deep energy based models could be a fruitful direction.

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

# A  DERIVATIONS

## A.1  MARGINAL PROBABILITY DISTRIBUTION OF VISIBLE UNITS OF GRBMS

We derive the marginal distribution of visible units as follows,

$$
\begin{aligned}
p(\mathbf{v}) &= \sum_{\mathbf{h}} p(\mathbf{v}, \mathbf{h}) \\
&= \frac{1}{Z} \sum_{\mathbf{h}} \exp(-E_\theta(\mathbf{v}, \mathbf{h})) \\
&= \frac{1}{Z} \exp\left(-\frac{1}{2}\left(\frac{\mathbf{v}-\boldsymbol{\mu}}{\boldsymbol{\sigma}}\right)^\top \left(\frac{\mathbf{v}-\boldsymbol{\mu}}{\boldsymbol{\sigma}}\right)\right) \sum_{\mathbf{h}} \exp\left(\left(\frac{\mathbf{v}}{\boldsymbol{\sigma}^2}\right)^\top W\mathbf{h} + \mathbf{b}^\top \mathbf{h}\right) \\
&= \frac{1}{Z} \exp\left(-\frac{1}{2}\left(\frac{\mathbf{v}-\boldsymbol{\mu}}{\boldsymbol{\sigma}}\right)^\top \left(\frac{\mathbf{v}-\boldsymbol{\mu}}{\boldsymbol{\sigma}}\right)\right) \prod_i \left(1 + \exp\left(\left(\left(\frac{\mathbf{v}}{\boldsymbol{\sigma}^2}\right)^\top W\right)_i + \mathbf{b}_i\right)\right) \\
&= \frac{1}{Z} \exp\left(-\frac{1}{2}\left(\frac{\mathbf{v}-\boldsymbol{\mu}}{\boldsymbol{\sigma}}\right)^\top \left(\frac{\mathbf{v}-\boldsymbol{\mu}}{\boldsymbol{\sigma}}\right)\right) \prod_i \exp\left(\mathrm{Softplus}\left(\left(\left(\frac{\mathbf{v}}{\boldsymbol{\sigma}^2}\right)^\top W\right)_i + \mathbf{b}_i\right)\right) \\
&= \frac{1}{Z} \exp\left(-\frac{1}{2}\left(\frac{\mathbf{v}-\boldsymbol{\mu}}{\boldsymbol{\sigma}}\right)^\top \left(\frac{\mathbf{v}-\boldsymbol{\mu}}{\boldsymbol{\sigma}}\right)\right) \exp\left(\mathrm{Softplus}\left(W^\top \frac{\mathbf{v}}{\boldsymbol{\sigma}^2} + \mathbf{b}\right)^\top \mathbf{1}\right) \\
&= \frac{1}{Z} \exp\left(-\frac{1}{2}\left(\frac{\mathbf{v}-\boldsymbol{\mu}}{\boldsymbol{\sigma}}\right)^\top \left(\frac{\mathbf{v}-\boldsymbol{\mu}}{\boldsymbol{\sigma}}\right) + \mathrm{Softplus}\left(W^\top \frac{\mathbf{v}}{\boldsymbol{\sigma}^2} + \mathbf{b}\right)^\top \mathbf{1}\right).
\end{aligned}
\tag{20}
$$

## A.2  GIBBS SAMPLING

---

**Algorithm 5** Gibbs Sampling for GRBMs

---

1: **Input**: number of steps $T$, burn-in step $\tilde{T}$
2: $\mathbf{v}^{(0)} \sim \mathcal{N}(\mathbf{0}, I)$
3: **For** $t = 1, \ldots, T$
4:     $\mathbf{h}^{(t)} \sim p(\mathbf{h}|\mathbf{v}^{(t-1)})$  ▷ following Eq. (4)
5:     $\mathbf{v}^{(t)} \sim p(\mathbf{v}|\mathbf{h}^{(t)})$  ▷ following Eq. (3)
6: **Return**: $\{(\mathbf{v}^{(t)}, \mathbf{h}^{(t)})|t = \tilde{T}+1, \cdots, T\}$

---

Gibbs sampling (Geman & Geman, 1984) is perhaps the most popular approach due to its simplicity. In the context of GRBMs, we can alternate between sampling hidden units given visible units and sampling visible units given hidden units. Alg. 5 is a blocked Gibbs sampler; it samples all visible units (a block of random variables) at once given all hidden units (the other block) and vice versa. Given the conditional independence in the bipartite graphical model, this block Gibbs sampler is equivalent to a univariate Gibbs sampler that updates one variable at a time given the others following some schedule. In fact, any schedule comprising a sequence of all hidden units followed by all visible units or vice versa would make the equivalence hold. In other words, it preserves the convergence of the original univariate Gibbs sampler and runs as fast as a blocked Gibbs sampler. Relying on this Gibbs sampler, we can get samples of visible and hidden units from the joint distribution in Eq. (2). We can then discard the samples within the burn-in stage and treat remaining ones as the final set of samples.

## A.3  LANGEVIN SAMPLING

The gradient of the marginal energy w.r.t. visible units is,

$$
\frac{\partial \tilde{E}(\mathbf{v})}{\partial \mathbf{v}} = \frac{\mathbf{v}-\boldsymbol{\mu}}{\boldsymbol{\sigma}^2} - \frac{W\,\mathrm{Sigmoid}\left(W^\top \frac{\mathbf{v}}{\boldsymbol{\sigma}^2} + \mathbf{b}\right)}{\boldsymbol{\sigma}^2}.
\tag{21}
$$

The cosine scheduler for annealing the step size is,

$$\alpha_k = \text{CosineScheduler}(k, K, \alpha_0) = \frac{1}{2}\alpha_0 \left(1 + \cos\left(\frac{k}{K}\pi\right)\right) \tag{22}$$

where $\alpha_k$ is the $k$-th step size, $\alpha_0$ is the initial step size, and $K$ is the total number of steps.

The derivation of the Metropolis adjustment for Langevin sampling is as follows,

$$\tilde{A}(\mathbf{v}^{(t)}, \mathbf{v}^{(t-1)}) = \min\left(1, \frac{p(\mathbf{v}^{(t)})q(\mathbf{v}^{(t-1)}|\mathbf{v}^{(t)})}{p(\mathbf{v}^{(t-1)})q(\mathbf{v}^{(t)}|\mathbf{v}^{(t-1)})}\right)$$

$$= \min\left(1, \frac{\exp\left(-\tilde{E}_\theta(\mathbf{v}^{(t)})\right)\exp\left(-\frac{1}{4\alpha_t}\left\|\mathbf{v}^{(t-1)} - \mathbf{v}^{(t)} + \alpha_t\frac{\partial\tilde{E}(\mathbf{v}^{(t)})}{\partial\mathbf{v}}\right\|^2\right)}{\exp\left(-\tilde{E}_\theta(\mathbf{v}^{(t-1)})\right)\exp\left(-\frac{1}{4\alpha_t}\left\|\mathbf{v}^{(t)} - \mathbf{v}^{(t-1)} + \alpha_t\frac{\partial\tilde{E}(\mathbf{v}^{(t-1)})}{\partial\mathbf{v}}\right\|^2\right)}\right)$$

$$= \min\left(1, \frac{\exp\left(-\tilde{E}_\theta(\mathbf{v}^{(t)}) - \frac{1}{4\alpha_t}\left\|\mathbf{v}^{(t-1)} - \mathbf{v}^{(t)} + \alpha_t\frac{\partial\tilde{E}(\mathbf{v}^{(t)})}{\partial\mathbf{v}}\right\|^2\right)}{\exp\left(-\tilde{E}_\theta(\mathbf{v}^{(t-1)}) - \frac{1}{4\alpha_t}\left\|\mathbf{v}^{(t)} - \mathbf{v}^{(t-1)} + \alpha_t\frac{\partial\tilde{E}(\mathbf{v}^{(t-1)})}{\partial\mathbf{v}}\right\|^2\right)}\right). \tag{23}$$

### A.4 GIBBS-LANGEVIN SAMPLING

We now derive the Metropolis Adjustment for Gibbs-Langevin sampling. At time step t-1, the proposal distribution in Alg. 2 is

$$q(\mathbf{v}, \mathbf{h}|\mathbf{v}^{(t-1)}, \mathbf{h}^{(t-1)}) = q(\mathbf{h}|\mathbf{v})q(\mathbf{v}|\mathbf{v}^{(t-1)}, \mathbf{h}^{(t-1)}), \tag{24}$$

where

$$q(\mathbf{h}|\mathbf{v}) = \text{Sigmoid}\left(W^\top\left(\frac{\mathbf{v}}{\boldsymbol{\sigma}^2}\right) + \mathbf{b}\right). \tag{25}$$

Denoting $\mathbf{v}^{(t-1)} = \tilde{\mathbf{v}}^{(0)}$ and $\mathbf{v} = \tilde{\mathbf{v}}^{(K)}$, we have,

$$q(\mathbf{v}|\mathbf{v}^{(t-1)}, \mathbf{h}^{(t-1)}) = q(\tilde{\mathbf{v}}^{(K)}|\tilde{\mathbf{v}}^{(0)}, \mathbf{h}^{(t-1)})$$

$$= \int \cdots \int \left(\prod_{k=1}^{K} q(\tilde{\mathbf{v}}^{(k)}|\tilde{\mathbf{v}}^{(k-1)}, \mathbf{h}^{(t-1)})\right) d\tilde{\mathbf{v}}^{(1)}\cdots d\tilde{\mathbf{v}}^{(K-1)}, \tag{26}$$

where

$$q(\mathbf{v}|\tilde{\mathbf{v}}^{(k-1)}, \mathbf{h}^{(t-1)}) = \mathcal{N}\left(\mathbf{v}\middle|\tilde{\mathbf{v}}^{(k-1)} - \alpha_k\frac{\partial E(\tilde{\mathbf{v}}^{(k-1)}, \mathbf{h}^{(t-1)})}{\partial\mathbf{v}}, 2\alpha_k I\right) \tag{27}$$

$$\frac{\partial E(\mathbf{v}, \mathbf{h})}{\partial\mathbf{v}} = \frac{\mathbf{v} - \boldsymbol{\mu} - W\mathbf{h}}{\boldsymbol{\sigma}^2}. \tag{28}$$

The key question here is how to derive the analytical form of $q(\tilde{\mathbf{v}}^{(K)}|\tilde{\mathbf{v}}^{(0)}, \mathbf{h}^{(t-1)})$. The most straightforward way is to compute the multiple integral directly. By fixing all variables except for $\tilde{\mathbf{v}}^{(k)}$ in Eq. (8), we can integrate out $\tilde{\mathbf{v}}^{(k)}$ analytically via the Gaussian integral trick, *i.e.*, $\int_{-\infty}^{\infty}\exp(-ax^2 + bx + c)dx = \sqrt{\frac{\pi}{a}}\exp(\frac{b^2}{4a} + c)$. Then by applying the same trick recursively, one can ideally integrate out all $\tilde{\mathbf{v}}^{(1)}, \ldots, \tilde{\mathbf{v}}^{(K-1)}$ in an analytical manner. However, this process is quite involved due to the fact that the integral of $\tilde{\mathbf{v}}^{(k)}$ depends on both $\tilde{\mathbf{v}}^{(k+1)}$ and $\tilde{\mathbf{v}}^{(k-1)}$.

We instead resort to the reparameterization trick. In particular, at the outer loop step $t$, the $k$-th inner loop step of Langevin sampling is as follows,

$$\tilde{\mathbf{v}}^{(k)} = \tilde{\mathbf{v}}^{(k-1)} - \alpha_k\frac{\partial E(\tilde{\mathbf{v}}^{(k-1)}, \mathbf{h}^{(t-1)})}{\partial\mathbf{v}} + \sqrt{2\alpha_k}\boldsymbol{\xi}_k$$

$$= \tilde{\mathbf{v}}^{(k-1)} - \alpha_k\frac{\tilde{\mathbf{v}}^{(k-1)} - \boldsymbol{\mu} - W\mathbf{h}^{(t-1)}}{\boldsymbol{\sigma}^2} + \sqrt{2\alpha_k}\boldsymbol{\xi}_k$$

$$= \left(1 - \frac{\alpha_k}{\boldsymbol{\sigma}^2}\right)\tilde{\mathbf{v}}^{(k-1)} + \alpha_k\frac{\boldsymbol{\mu} + W\mathbf{h}^{(t-1)}}{\boldsymbol{\sigma}^2} + \sqrt{2\alpha_k}\boldsymbol{\xi}_k, \tag{29}$$

where $\forall k \in \{1, \ldots, K\}$, $\boldsymbol{\xi}_k \sim \mathcal{N}(0, I)$. This discretization of Langevin dynamics gives a sample path of the distribution $q(\tilde{\mathbf{v}}^{(K)}|\tilde{\mathbf{v}}^{(0)}, \mathbf{h}^{(t-1)})$. We now show that this sample path could be reparameterized as a simpler one which gives the desirable analytical form of $q(\tilde{\mathbf{v}}^{(K)}|\tilde{\mathbf{v}}^{(0)}, \mathbf{h}^{(t-1)})$. To simplify the derivation, we introduce $\boldsymbol{\beta}_k = \prod_{j=k+1}^{K} \left(1 - \frac{\alpha_j}{\boldsymbol{\sigma}^2}\right)$, $\forall k \in \{0, \ldots, K-1\}$ and $\boldsymbol{\beta}_K = \mathbf{1}$. Therefore, after $K$ steps, we have,

$$
\begin{aligned}
\tilde{\mathbf{v}}^{(K)} &= \left(1 - \frac{\alpha_K}{\boldsymbol{\sigma}^2}\right) \tilde{\mathbf{v}}^{(K-1)} + \alpha_K \frac{\boldsymbol{\mu} + W\mathbf{h}^{(t-1)}}{\boldsymbol{\sigma}^2} + \sqrt{2\alpha_K}\boldsymbol{\xi}_K \\
&= \left(\prod_{k=1}^{K}\left(1 - \frac{\alpha_k}{\boldsymbol{\sigma}^2}\right)\right)\tilde{\mathbf{v}}^{(0)} + \sum_{k=1}^{K}\left(\prod_{j=k+1}^{K}\left(1 - \frac{\alpha_j}{\boldsymbol{\sigma}^2}\right)\right)\left(\alpha_k\frac{\boldsymbol{\mu}+W\mathbf{h}^{(t-1)}}{\boldsymbol{\sigma}^2} + \sqrt{2\alpha_k}\boldsymbol{\xi}_k\right) \\
&= \boldsymbol{\beta}_0\tilde{\mathbf{v}}^{(0)} + \left(\sum_{k=1}^{K}\boldsymbol{\beta}_k\alpha_k\right)\frac{\boldsymbol{\mu}+W\mathbf{h}^{(t-1)}}{\boldsymbol{\sigma}^2} + \sum_{k=1}^{K}\boldsymbol{\beta}_k\sqrt{2\alpha_k}\boldsymbol{\xi}_k
\end{aligned}
\tag{30}
$$

Here $\{\boldsymbol{\xi}_k|k = 1, \ldots, K\}$ are independent random variables from the standard Normal distribution. Since we know that the linear combination of several independent Gaussian random variables leads to another Gaussian random variable, we have

$$
q(\tilde{\mathbf{v}}^{(K)}|\tilde{\mathbf{v}}^{(0)}, \mathbf{h}^{(t-1)}) = \mathcal{N}\left(\boldsymbol{\beta}_0\tilde{\mathbf{v}}^{(0)} + \left(\sum_{k=1}^{K}\boldsymbol{\beta}_k\alpha_k\right)\frac{\boldsymbol{\mu}+W\mathbf{h}^{(t-1)}}{\boldsymbol{\sigma}^2}, \sum_{k=1}^{K}2\alpha_k\boldsymbol{\beta}_k^2\right).
\tag{31}
$$

We can compute the acceptance probability,

$$
\begin{aligned}
&A((\mathbf{v}^{(t)}, \mathbf{h}^{(t)}), (\mathbf{v}^{(t-1)}, \mathbf{h}^{(t-1)})) \\
&= \min\left(1, \frac{p(\mathbf{v}^{(t)}, \mathbf{h}^{(t)})q(\mathbf{v}^{(t-1)}, \mathbf{h}^{(t-1)}|\mathbf{v}^{(t)}, \mathbf{h}^{(t)})}{p(\mathbf{v}^{(t-1)}, \mathbf{h}^{(t-1)})q(\mathbf{v}^{(t)}, \mathbf{h}^{(t)}|\mathbf{v}^{(t-1)}, \mathbf{h}^{(t-1)})}\right) \\
&= \min\left(1, \frac{p(\mathbf{v}^{(t)}, \mathbf{h}^{(t)})q(\mathbf{h}^{(t-1)}|\mathbf{v}^{(t-1)})q(\mathbf{v}^{(t-1)}|\mathbf{v}^{(t)}, \mathbf{h}^{(t)})}{p(\mathbf{v}^{(t-1)}, \mathbf{h}^{(t-1)})q(\mathbf{h}^{(t)}|\mathbf{v}^{(t)})q(\mathbf{v}^{(t)}|\mathbf{v}^{(t-1)}, \mathbf{h}^{(t-1)})}\right) \\
&= \min\left(1, \frac{\exp\left(-E_\theta(\mathbf{v}^{(t)}, \mathbf{h}^{(t)}) - \left\|\frac{\mathbf{v}^{(t-1)}-\boldsymbol{\beta}_0\mathbf{v}^{(t)}-\boldsymbol{a}(\boldsymbol{\mu}+W\mathbf{h}^{(t)})}{\sqrt{2}\tilde{\boldsymbol{\sigma}}}\right\|^2\right)q(\mathbf{h}^{(t-1)}|\mathbf{v}^{(t-1)})}{\exp\left(-E_\theta(\mathbf{v}^{(t-1)}, \mathbf{h}^{(t-1)}) - \left\|\frac{\mathbf{v}^{(t)}-\boldsymbol{\beta}_0\mathbf{v}^{(t-1)}-\boldsymbol{a}(\boldsymbol{\mu}+W\mathbf{h}^{(t-1)})}{\sqrt{2}\tilde{\boldsymbol{\sigma}}}\right\|^2\right)q(\mathbf{h}^{(t)}|\mathbf{v}^{(t)})}\right),
\end{aligned}
\tag{32}
$$

where $q(\mathbf{h}_j = 1|\mathbf{v}) = \left[\text{Sigmoid}\left(W^\top\frac{\mathbf{v}}{\boldsymbol{\sigma}^2} + \mathbf{b}\right)\right]_j$, $\boldsymbol{a} = \frac{\sum_{k=1}^{K}\boldsymbol{\beta}_k\alpha_k}{\boldsymbol{\sigma}^2}$, and $\tilde{\boldsymbol{\sigma}}^2 = \sum_{k=1}^{K}2\alpha_k\boldsymbol{\beta}_k^2$.

## A.5 LEARNING

We derive the detailed gradients of the marginalized log likelihood of visible units w.r.t. model parameters as below.

$$
\nabla W_{ij} = \left\langle \frac{\mathbf{v}_i}{\boldsymbol{\sigma}_i^2}\left[\text{Sigmoid}\left(W^\top\frac{\mathbf{v}}{\sigma^2} + \mathbf{b}\right)\right]_j\right\rangle_d - \left\langle \frac{\mathbf{v}_i}{\boldsymbol{\sigma}_i^2}\left[\text{Sigmoid}\left(W^\top\frac{\mathbf{v}}{\sigma^2} + \mathbf{b}\right)\right]_j\right\rangle_m
\tag{33}
$$

$$
\nabla\mu_i = \left\langle \frac{\mathbf{v}_i - \mu_i}{\boldsymbol{\sigma}_i^2}\right\rangle_d - \left\langle \frac{\mathbf{v}_i - \mu_i}{\boldsymbol{\sigma}_i^2}\right\rangle_m
\tag{34}
$$

$$
\begin{aligned}
\nabla\log\boldsymbol{\sigma}_i^2 &= \left\langle \frac{(\mathbf{v}_i - \mu_i)^2}{2\boldsymbol{\sigma}_i^2} - \frac{\sum_j \mathbf{v}_i W_{ij}\left[\text{Sigmoid}\left(W^\top\frac{\mathbf{v}}{\sigma^2} + \mathbf{b}\right)\right]_j}{\boldsymbol{\sigma}_i^2}\right\rangle_d \\
&\quad - \left\langle \frac{(\mathbf{v}_i - \mu_i)^2}{2\boldsymbol{\sigma}_i^2} - \frac{\sum_j \mathbf{v}_i W_{ij}\left[\text{Sigmoid}\left(W^\top\frac{\mathbf{v}}{\sigma^2} + \mathbf{b}\right)\right]_j}{\boldsymbol{\sigma}_i^2}\right\rangle_m
\end{aligned}
\tag{35}
$$

$$
\nabla\mathbf{b}_i = \left\langle\left[\text{Sigmoid}\left(W^\top\frac{\mathbf{v}}{\sigma^2} + \mathbf{b}\right)\right]_i\right\rangle_d - \left\langle\left[\text{Sigmoid}\left(W^\top\frac{\mathbf{v}}{\sigma^2} + \mathbf{b}\right)\right]_i\right\rangle_m.
\tag{36}
$$

## B  MORE EXPERIMENTAL RESULTS

For all experiments, we set the initial variances of GRBMs to be $1$, clip the gradient norm to be no larger than $10$, and use SGD with neither momentum nor weight decay. We divide the total energy of a mini-batch by the batch size so that we are minimizing the average negative log likelihood. We also decay the learning rate of SGD from the initial value $0.01$ to $0$ using the same cosine scheduler as described in Eq. (22). The burn-in step in CD learning is set to $0$, *i.e.*, we do not discard any samples from any Markov chains. For all experiments involving images, we standardize the input image by subtracting the pixel-wise mean and dividing by the pixel-wise standard deviation. For color images, the subtraction and division is performed channel-wise. The details of all baselines on MNIST are as follows.

- **VAE.** We use an encoder with $4$ convolutional blocks ($3 \times 3$ Conv+BN+ReLU) along with a 2-layer MLP. The decoder contains a 2-layer MLP followed by a $4$ convolutional block ($3 \times 3$ Conv+BN+ReLU) and a 1-hidden-layer MLP. The total number of parameters is around 3.98M.

- **2sVAE.** For the encoder, we use $4$ convolutional blocks ($3 \times 3$ Conv+BN+ReLU). For the decoder, we use $2$ convolutional blocks ($3 \times 3$ Conv+BN+ReLU) followed by another $2$ convolutional blocks ($3 \times 3$ ConvTranspose+BN+ReLU). The total number of parameters is around 13.58M.

- **PixelCNN++.** We have $8$ masked convolutional blocks ($3 \times 3$ Conv+BN+ReLU) which result in 1.41M parameters.

- **WGAN.** For the discriminator, we have $4$ convolutional blocks ($3 \times 3$ Conv+BN+ReLU). For the generator, we have $4$ convolutional blocks ($3 \times 3$ ConvTranspose+BN+ReLU). The total number of parameters is around 1.73M.

- **NVAE.** We have $44$ ResBlock and $42$ ResBlock for the encoder and decoder respectively, which results in around 33.36M parameters.

Our GRBMs instead have a single hidden layer and about 3.21M parameters.

### B.1  GAUSSIAN MIXTURE DENSITIES

The batch size and the hidden size are set to $100$ and $256$ respectively. We adjust at every step whenever Metropolis adjustmentment is used as the experiments with Gaussian mixture densities are fast. Although smaller hidden size could work, but we found this size makes learning converges stably for all sampling algorithms. To ensure a fair comparison, we train all GRBMs for 50K epochs and use the last model to draw the density plots and samples, despite the learning processes with most of inference algorithms converge within 5K to 10K epochs.

### B.2  ABLATION STUDY ON MNIST

In this part, we perform ablation study on MNIST to investigate the effect of several important factors. In all experiments on MNIST, we set batch size to $512$ and the number of epochs to $3000$. First, we vary the CD step and the hidden size while fixing the other hyperparameters. The results are shown in Table 2. We found that $4096$ hidden size and $100$ CD steps work the best on MNIST. More CD steps would potentially be better but take longer time to train. Then we turn to study the number of Langevin sampling steps, the adjust step size, the initial Langevin step size, and its annealing. Here annealing means we decay the initial Langevin step size to $0$ following the cosine scheduler as training goes on. The results are shown in Table 3. We can see that the larger the initial Langevin step size, the better the performance. But values larger than $0.04$ would sometimes make the sampling numerically fail. The more the Langevin steps, the better the performance would be. Again, it comes with more computational cost with more Langevin steps. We also find that it may not be necessary to adjust at every step and annealing the initial step size slightly improves the performance of Gbbis-Langevin with adjustment. We further study the effect of standard Normal noise vs. data in the initialization of negative Markov chains in CD. In particular, given a mini-batch of data, we randomly draw a portion of them so that their negative Markov chains start with samples drawn from the standard Normal distribution, same as what we did in modified CD. But for the

remaining portion of data, we start their negative Markov chains from the data, same as what we did for the positive Markov chains. This can help us understand the importance of noises in the initialization of modified CD. The results are shown in Table 4. It is clear that the sample quality decreases with more data being used as the initialization. This is expected since the discrepancy between the initial distributions of the negative Markov chain during training and the Markov chain during testing is increasing. Note that if we use data as initialization for all samples, then the underlying learning method reduces to the original CD.

## B.3 MORE VISUAL RESULTS

We train 3K epochs for experiments on both FashionMNIST and CelebA-32 datasets. For CeleA-2K-64, we train 4K epochs. The batch size on FashionMNIST and CelebA-32 is $512$ whereas the batch size on CeleA-2K-64 is $100$.

We show the samples drawn from the best GRBMs learned with different sampling methods in Fig. 5. It is clear that samples corresponding to Gibbs-Langevin have better visual qualities than those from Langevin and Gibbs. We also show more results of GRBMs learned with Gibbs-Langevin in Fig. 6, Fig. 7, Fig. 8, and Fig. 9.

| Methods | CD Step | Hidden Size | FID |
|---|---|---|---|
| Gibbs-Langevin wo. Adjust | 50 | 2048 | 32.33 |
| Gibbs-Langevin wo. Adjust | 50 | 4096 | 21.02 |
| Gibbs-Langevin wo. Adjust | 50 | 8192 | 22.05 |
| Gibbs-Langevin wo. Adjust | 100 | 4096 | **17.49** |

Table 2: Ablation study of the hidden size and the number of CD steps on MNIST dataset.

| Methods | Langevin Step $K$ | Langevin Step Size $\alpha_0$ | Anneal $\alpha_0$ | Adjust Step $\eta$ | FID |
|---|---|---|---|---|---|
| Gibbs-Langevin wo. Adjust | 1 | 20 | ✗ | - | 35.08 |
| Gibbs-Langevin wo. Adjust | 5 | 20 | ✗ | - | 19.00 |
| Gibbs-Langevin wo. Adjust | 10 | 20 | ✗ | - | **17.49** |
| Gibbs-Langevin wo. Adjust | 10 | 10 | ✗ | - | 21.05 |
| Gibbs-Langevin wo. Adjust | 10 | 5 | ✗ | - | 25.67 |
| Gibbs-Langevin w. Adjust | 10 | 20 | ✗ | 0 | 21.31 |
| Gibbs-Langevin w. Adjust | 10 | 20 | ✗ | 25 | 21.25 |
| Gibbs-Langevin w. Adjust | 10 | 20 | ✗ | 50 | 20.64 |
| Gibbs-Langevin w. Adjust | 10 | 20 | ✓ | 50 | 19.27 |

Table 3: Ablation study of the number of Langevin steps $K$, the initial Langevin step size $\alpha_0$, annealing of the initial Langevin step size, and the Metropolis adjust step $\eta$ on MNIST dataset. All runs use 100 CD steps.

| Data (%) | Noise (%) | FID |
|---|---|---|
| 0 | 100 | **17.49** |
| 25 | 75 | 19.02 |
| 50 | 50 | 21.03 |
| 75 | 25 | 53.20 |

Table 4: Ablation study of the initialization of negative Markov chains.

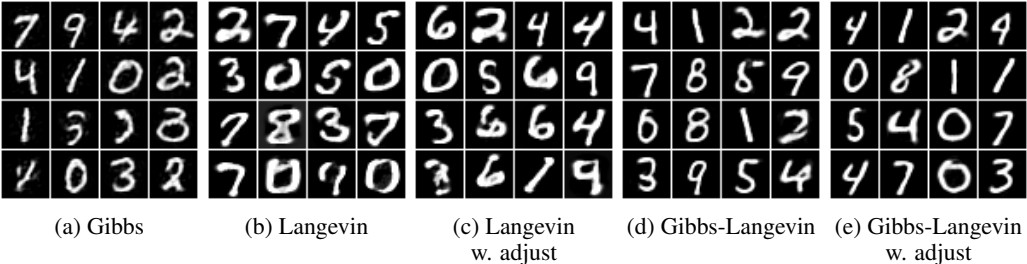

(a) Gibbs     (b) Langevin     (c) Langevin     (d) Gibbs-Langevin     (e) Gibbs-Langevin
w. adjust                  w. adjust

Figure 5: Samples from GRBMs learned with different sampling algorithms on MNIST.

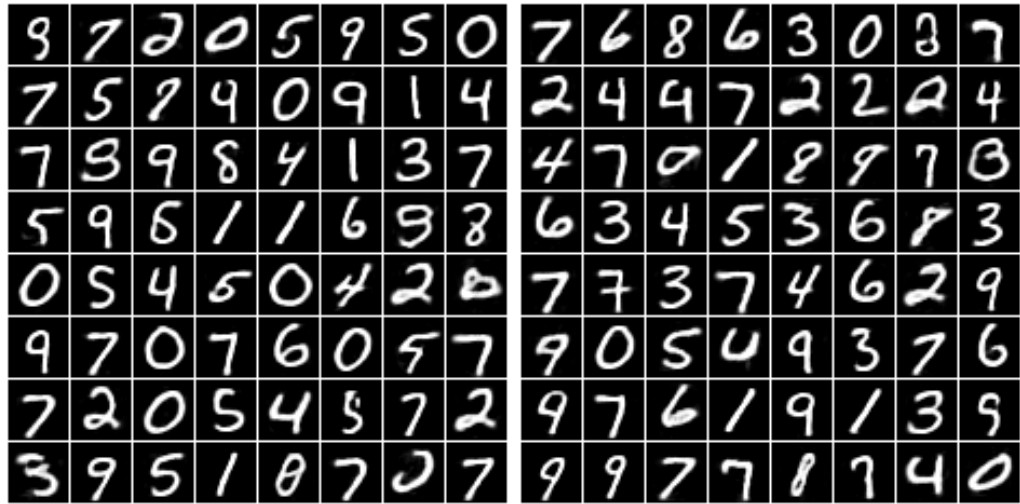

Figure 6: More samples from the learned GRBM (Gibbs-Langevin) on MNIST.

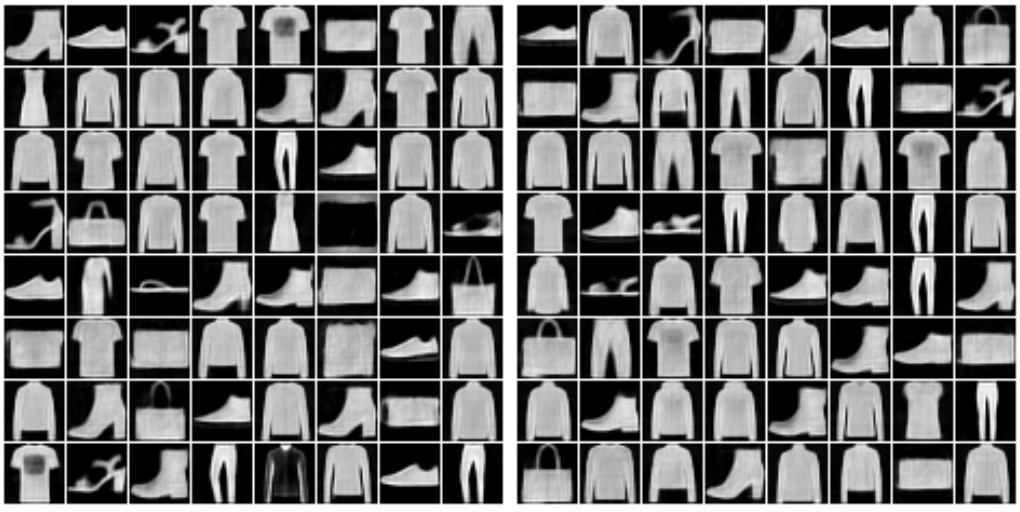

Figure 7: More samples from the learned GRBM (Gibbs-Langevin) on FashionMNIST.

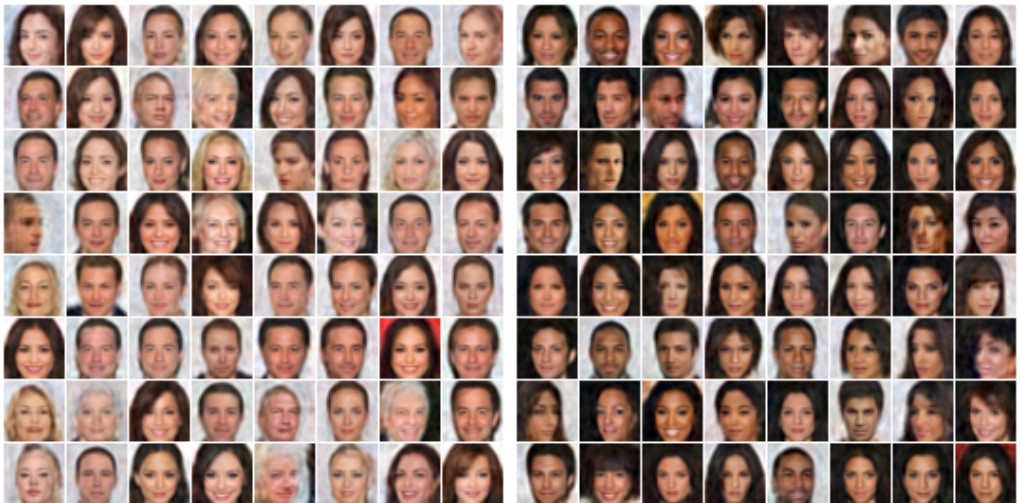

Figure 8: More samples from the learned GRBM (Gibbs-Langevin) on CelebA-32.

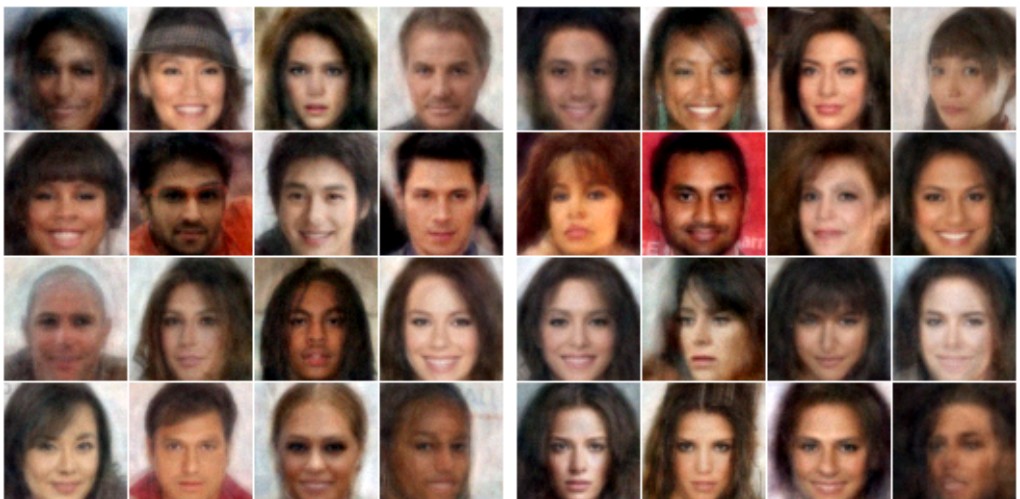

Figure 9: More samples from the learned GRBM (Gibbs-Langevin) on CelebA-2K-64.

