# OpenReview forum: "Gaussian-Bernoulli RBMs Without Tears"
_ICLR.cc/2023/Conference — Submitted to ICLR 2023_

### Official Review · Reviewer_sW5s · 2022-10-25

**Confidence:** 4
**Correctness:** 3
**Technical Novelty And Significance:** 3
**Empirical Novelty And Significance:** 2
**Recommendation:** 6

**Clarity, Quality, Novelty And Reproducibility:**

This paper is overall well-written and nicely polished. It makes a reasonable attempt to provide sufficient background for the readers to understand the proposed methods. One minor issue is that the font size in fig 1 is too tiny to read.

**Strength And Weaknesses:**

The main contribution of this work is the proposed Gibbs-Langevin sampling for GRBMs and the modified contrastive divergence that allows the GRBMs to generate images from noise and to be compared with other existing deep generative models. It is interesting to see GRBMS, models with relatively simple architectures, are capable of generating reasonable images. All the proposed methods are nicely explained and seem technically sound. Still, here are some concerns/confusions:
- I find the motivation on why using GRBMs for generative modeling and when they are preferable to other models is somewhat lacking. Even though the paper claims that an important motivation is that a GRBM can convert real-valued data to stochastic binary data, it is not something unique to GRBMs since a VAE with discrete latent space can do the same thing. In terms of empirical evaluations, the FID scores of GRBMs are still a lot worse than the existing generative models.
- In related work section, it says that the proposed learning algorithm allows to learn a much lower variances compared with other methods. I wonder why smaller learned variances are a good thing. Would they lead to a less diverse model distribution?
- In Section 3.1, paragraph Gibbs-Langevin sampling, the authors claim that it performs better than generic Gibbs due to the leverage of gradient information of log density. Similar comparison between Gibbs-Langevin and Langevin is missing. I wonder why in Table one it is much better than Langevin. Any intuition why.
- In experimental section, it is vague what does adjustment refer to. Do you mean gradient clipping, modified CD or something else? This greatly harms the readability of empirical results.

**Summary Of The Paper:**

This work aims to train Gausssian-Bermoulli restricted Boltzmann machines (GRBMs) for generative modeling. For inference, it proposes to use a hybrid sampling that combines Gibbs sampling steps with Langevin samplers for GRBMs instead of the vanilla Gibbs sampling such that the gradient information of the log density can be leveraged. For learning, a modified contrastive divergence is proposed that encourages the model to generate samples from noises. The proposed training framework is evaluated on both synthetic and real-world datasets: Gaussian Mixtures, MNIST, FashionMNIST, and CelebA.


**Summary Of The Review:**

Overall, nice improvements on challenging GRBMs.

---

> ### Author Response · Authors · 2022-11-18
> **Response To Reviewer sW5s**
>
> Thank you for your positive review comments!
>
> > **Q1: Motivation & Discrete VAEs.**
>
> A1: Please see our general response to the concern about motivation.
>
> We agree that discrete VAEs could serve a similar purpose of converting continuous data to discrete representations. To learn discrete VAEs, many gradient estimation techniques have been explored, e.g., straight-through, REINFORCE, and relaxed reparameterization (e.g., Gumbel-Softmax), which are quite different from contrastive divergence used in GRBMs. Therefore, from the perspective of learning techniques, we are exploring novel sampling methods for learning such hybrid models that have this continuous-to-discrete function.
>
> > **Q2: Why is learning smaller variances a good thing?**
>
> A2: Thanks for raising this good point! We clarified this point further in the revised paper.
>
> There are two types of variances related to GRBMs in modelling the density of images. 1) Intrinsic variance: the variance of image contents. 2) Extrinsic variance: the variance parameter of GRBM. Given the image dataset, the intrinsic variance is fixed and could be large or small depending on the dataset itself (e.g., the intrinsic variance in CIFAR10 is larger than MNIST).
> The extrinsic variance introduces artificial Gaussian noise to images generated by GRBMs since the conditional distribution of visible units conditioned on hidden ones is Gaussian. We typically do not observe Gaussian noise in images from these datasets and hope to generate clean images. Therefore, learning a small extrinsic variance is a good thing.
>
> > **Q3: Any intuition why Gibbs-Langevin is better than Langevin?**
>
> A3: Our intuition is as follows.
>
> First, there are many different joint distributions that share the same marginal distribution.
> For example, consider two joint distributions P1(X,Y) and P2(X,Y) below,
>
> P1(X=0,Y=0) = 1/6, P1(X=0,Y=1) = 2/6, P1(X=1,Y=0) = 2/6, P1(X=1,Y=1) = 1/6,
>
> P2(X=0,Y=0) = 2/6, P2(X=0,Y=1) = 1/6, P2(X=1,Y=0) = 1/6, P2(X=1,Y=1) = 2/6.
>
> One can check that P1(X=0) = P2(X=0) = P1(X=1) = P2(X=1) = 1/2.
> Therefore, there could exist multiple GRBMs with the same marginal distribution of visible units.
> From this perspective, directly using the marginal distribution and employing Langevin sampling and learning may not fully exploit the underlying bipartite structure of GRBMs.
>
> Second, the dynamics of Langevin sampling in the space of visible units are more or less smooth as long as the noise level and the step size are reasonable, whereas the dynamics of Gibbs-Langevin in the space of visible units could have large jumps due to the switches of binary hidden units. Therefore, the alternative sampling between continuous and discrete random variables in Gibbs-Langevin could potentially explore more space than Langevin.
>
> > **Q4: In experimental section, it is vague what does adjustment refer to. Do you mean gradient clipping, modified CD or something else? This greatly harms the readability of empirical results.**
>
> A4: Adjustment refers to Metropolis adjustment. We have clarified this in the revised paper.

---

### Official Review · Reviewer_SJ4K · 2022-10-25

**Confidence:** 3
**Correctness:** 4
**Technical Novelty And Significance:** 4
**Empirical Novelty And Significance:** 3
**Recommendation:** 6

**Clarity, Quality, Novelty And Reproducibility:**

- Quality: The paper was high quality and clear.
- Novelty: The techniques and theoretical insights provided in the paper were novel, and significantly advance the potential for the practical usefulness of GRBMs.
- Reproducibility: No code was attached with the submission.


**Strength And Weaknesses:**

Strengths:
- The theoretical insights in the paper were sound and important.
- This is also the first time that GRBMs can generate images unconditionally. Although the sample qualities can be improved, the fact that this could be done is impressive even with a small model architecture is impressive.

Weaknesses:
- Although I don’t expect the FIDs for samples generated from other datasets (e.g. CelebA) to be competitive, would the authors comment on the gap between GRBMs and existing (maybe simple) baselines?
- It would have also been nice to have additional experimental details (e.g. does the VAE architecture match that of the GRBM?).


**Summary Of The Paper:**

This paper introduces many new insights that ease the training of and push the capabilities of Gaussian-Bernoulli Restricted Boltzmann Machines (GRBMs). In particular, they propose a hybrid Gibbs-Langevin sampling algorithm for inference, as well as a modified CD algorithm using two Markov chains (paired with gradient clipping) that allows for training GRBMs without many of the empirical hacks necessary to stabilize training in practice. The authors demonstrate the effectiveness of their approach on synthetic and benchmark image datasets (e.g. MNIST, CelebA).

**Summary Of The Review:**

The paper provides novel and interesting theoretical insights improving the performance of GRBMs in practice, as well as several proof of concept experiments for how they can be leveraged for generative modeling. This paper will be of interest to the community, and I imagine this encouraging new follow-up works.

---

> ### Author Response · Authors · 2022-11-18
> **Response To Reviewer SJ4K**
>
> Thank you for your positive review comments!
>
> > **Q1: Comment on the gap between GRBMs and existing baselines.**
>
> A1: The gaps w.r.t. baselines on more challenging datasets like CelebA are still significant. We attribute the gap to the shallow architecture of our GRBM. As discussed in the motivation portion of our response, a deep Boltzmann machine (DBM) with the 1st layer being a GRBM to convert continuous data to binary may be able to close  the gap thanks to its deep architecture. But inference and learning will be more intricate, as the conditional independence structure of GRBM goes away in DBM. Greedy layer-wise pre-training could be a promising direction.
>
> On the other hand, we found GRBMs tend to memorize the data by learning more filters that capture global patterns of images. Therefore, exploring convolutional GRBMs and their DBM counterparts would potentially enable the learning of more localized filters. This could help improve the sample quality, just as ConvVAEs work better than MLP-based VAEs.
>
> > **Q2: Additional experimental details like architecture of baselines.**
>
> A2: On MNIST, all baselines except for NVAE have roughly 8 layers (VAEs have a 4-layer convolutional encoder and a 4-layer convolutional decoder). NVAE has a 44-resbock encoder and a 42-layer decoder. The number of parameters of VAE, 2sVAE, PixelCNN++, WGAN, and NVAE are roughly 3.98e+6, 1.35e+7, 1.41e+6, 1.73e+6, and 3.33e+7 respectively. Our GRBMs instead have 1 layer and about 3.21e+6 parameters. We added more details in the revised appendix.

---

> > ### Comment · Reviewer_SJ4K · 2022-12-13
> > **Thank you**
> >
> > I'd like to thank the authors for their response. Upon reading and discussing with the other reviewers, I've lowered my score a little bit. While I do think that the paper is still interesting, it's not clear that image generation is the best application for this kind of work. It would make the paper much stronger to reframe the story (and provide results) in terms of model interpretability + uncertainty quantification, rather than to try to compete with existing models (e.g. EBMs) with image generation.

---

### Official Review · Reviewer_Uikc · 2022-10-25

**Confidence:** 5
**Correctness:** 4
**Technical Novelty And Significance:** 2
**Empirical Novelty And Significance:** 2
**Recommendation:** 3

**Clarity, Quality, Novelty And Reproducibility:**

The novelty of this work is somewhat lacking because it uses an established energy function and established methods such as Langevin sampling and noise-initialized MCMC from the EBM literature. The quality of results are good for the RBM family but very limited compared to virtually all contemporary generative model families.

**Strength And Weaknesses:**

STRENGTHS:

1. The work improves on GRBM learning and introduces the first RBM method that can generate images from scratch.
2. Sample quality is reasonable given the restrictions of the potential energy.

WEAKNESSES:

1. The motivation of using GRBMs is unclear. From one perspective, the visible GRBM is simply a deep EBM with a restricted architecture. The hidden units distinguish the GRBM and RBM family, but the applications do not depend on the hidden representations. A clear motivation for the use of a GRBM would greatly strengthen the paper.
2. The presentation of the modified CD initialization is missing reference to related EBM works such as [1] that also use noise-initialized MCMC for both training and testing. It might be useful to perform an ablation study to investigate the effect of the proportion of data samples and noise initialized samples. It is possible that the data samples might not be needed at all.

[1] Learning Non-Convergent Non-Persistent Short-Run MCMC Toward Energy-Based Model
https://arxiv.org/pdf/1904.09770.pdf

**Summary Of The Paper:**

This work revisits the GRBM to improve its learning and sampling process. One key innovation is introducing Langevin sampling to GRBMs. Two variants are proposed: sampling the visible marginal potential, and hybrid Gibbs-Langevin sampling that trades off between Gibbs updates of hidden units and Langevin updates of visible units. The other key innovation is initializing MCMC samples from noise so that the model can generate samples from noise once training is over. Experiments show that the learning method can successfully train GRBMs that can generate reasonable images from scratch.

**Summary Of The Review:**

The work shows improvements within the GRBM family, but this potential is known to be very restrictive and even in ideal circumstances will likely be unable to match a wide variety of contemporary generative models. Without a clear motivation for using the GRBM, I recommend against accepting.

---

> ### Author Response · Authors · 2022-11-18
> **Response to Reviewer Uikc**
>
> Thank you for your helpful review comments!
>
> > **Q1: Motivation**
>
> A1: Please see our general response to the concern about motivation.
>
>
> > **Q2: Missing reference about modified CD**
>
> A2: Thanks for pointing out the reference! It is indeed relevant. We have cited and discussed it in the revised paper. But we also want to highlight some differences.
>
> First, we are primarily focusing on GRBMs, which are hybrid (discrete hidden + continuous visible units) EBMs, whereas [1] focuses on continuous EBMs. Therefore, the sampling methods are quite different. We exploit hybrid samplers like Gibbs-Langevin, whereas [1] directly adopts Langevin.
>
> Second, since GRBMs are hybrid, our modified CD only randomly draws visible units (a subset of all variables) from standard Normal and starts the Gibbs-style alternative sampling, whereas the method in [1] draws all variables at once from a uniform distribution with range [-1,1] and starts the Langevin sampling as it focuses on EBMs with only continuous random variables.
>
> Third, the method in [1] adds additional Gaussian noise to clean data in the positive phase to smooth the data distribution, whereas our modified CD does not incorporate this; we found it does not yield any performance gain for GRBMs but it introduces more hyperparameters.
>
>
> > **Q3: The novelty of this work is somewhat lacking because it uses an established energy function and established methods such as Langevin sampling and noise-initialized MCMC from the EBM literature.**
>
> A3: We’d like to clarify that our work is novel in at least two respects.
>
> First, many works involving EBMs, especially recent ones on deep EBMs, focus exclusively on continuous random variables where applying Langevin sampling is natural and straightforward. However, we deal with hybrid graphical models which consist of both continuous and discrete random variables. It is non-trivial to design effective hybrid samplers. Furthermore, the Gibbs sampler, which is the predominant sampler in the GRBM literature, is already hybrid and permits exact sampling at individual alternative steps. It is not obvious from previous work that  Langevin sampling would still provide improvements in this context. Our work broadens the area where Langevin sampling could be effective.
>
> Second, to the best of our knowledge, our work is the first to show that GRBMs, despite being a simple hybrid EBM, can generate good images that are comparable to those of modern generative models.
>
>
> > **Q4: Ablation study to investigate the effect of the proportion of data samples and noise initialized samples.**
>
> A4: Thanks for pointing this out. We recently performed an ablation study about this effect on MNIST. We control all hyperparameters except for the initialization of negative Markov chains to be the same, e.g., the sampling method is Gibbs-Langevin without adjustment and the hidden dimension is 4096. We randomly draw a portion of samples within the minibatch to start their negative Markov chain from noise, whereas the remaining ones start from the data. The results are summarized below.
>
> | Data Percentage | Noise Percentage | FID |
> |:--------------------:|:-------------:|------:|
> |        0% |100% |  17.49 |
> | 25% |75%|19.02|
> |50%|50%|21.03|
> |75%|25%|53.20|
>
>
> It is clear that with more negative Markov chains initialized from the data, the sample quality gets worse. This is expected since the discrepancy between the initial distributions of the negative Markov chain during training and the Markov chain during testing is increasing. We added this result in the revised appendix.

---

> > ### Comment · Reviewer_Uikc · 2022-12-05
> > **Thanks for the response. Paper would be strengthened with a focused application.**
> >
> > Thanks to authors for their thorough response. I agree that GRBM/DBM learning has the potential to lead to models that have capabilities beyond those of the continuous and fully visible EBM, and that this work provides a good starting point for such an investigation. While there are some differences between EBM learning the proposed GRBM method (the most notable to me being the alternate sampling between the visible and hidden units), I personally did not find the method or results surprising or highly novel given related developments in the EBM literature. While this paper shows definite progress in the area of GRBMs, I personally still do not find the results novel or strong enough for ICLR. In the general response, the authors bring up some areas for future study such as DBM learning and quantum computing applications where GRBM learning is crucial and continuous EBM learning would not suffice. In my opinion, a strong application in this direction is the missing ingredient that would offer a clear and compelling use case for the proposed methods and help this work achieve maximum impact. I applaud the efforts made by the authors and hope that this work can be the foundation for future efforts in this direction. In its present state, I do not this the work is ready for publication.

---

### Author Response · Authors · 2022-11-18
**General Response From Authors**

We thank all the reviewers for their constructive and valuable feedback. We have addressed most of the suggestions and updated our manuscript with edits colored in blue. We also include the code in the supplementary file. We respond to the common concern here and other questions individually.

We explain our motivation from the following perspectives.

1. **Foundation for Building Deep Boltzmann Machines**. Progress towards training better RBMs/GRBMs will facilitate research into Deep Boltzmann Machines (DBMs), which are more expressive than RBMs/GRBMs. DBMs are quite different from current deep energy based models (EBMs), since DBMs have hierarchically dependent binary latent variables, whereas EBMs often do not contain any latent variables. Having latent variables could help improve the model’s interpretability and quantify uncertainty. More importantly, compared to other generative models that are trained with backpropagation, the learning of deep Boltzmann machines typically leverages Gibbs-style alternative local sampling and local gradient computations, and are thus widely regarded to be more biologically plausible. Note that the predominant learning algorithm of deep EBMs (i.e., Langevin sampling-based Contrastive Divergence) still requires backpropagation to compute the gradient of the energy function w.r.t. observed random variables. And finally, since plain RBMs cannot deal with continuous data (e.g., images), our work on GRBMs enables one to effectively convert continuous data to binary latent codes, which can then be used as a front-end of DBMs for continuous data (images, video, speech, etc).

2. **Inference and Learning Algorithms for Hybrid EBMs**. Since GRBMs are perhaps one of the simplest hybrid (continuous observable random variables + discrete/binary latent variables) EBMs, investigating inference and learning algorithms for GRBMs provides a foundation for further research on general hybrid deep EBMs, which would be useful for many applications, e.g., generating (continuous) images and their (discrete) scene graphs. In particular, the sampling techniques (e.g., Gibbs-Langevin) we proposed are directly applicable to hybrid deep EBMs. As discussed in the conclusion, we leave the investigation of their effectiveness for hybrid deep EBMs as future work.

3. **Applications in Quantum Computing and Physics**. Although RBMs and GRBMs are mainly of historical interest in machine learning, they continue to be actively studied in physics and quantum computing [1,2,3]. Many quantum computing devices require the problem to be formulated as Ising models or binary optimization problems, which are naturally satisfied by RBMs and GRBMs. For example, in [1], a GRBM has been used to convert the continuous input signal to a binary one, which is further fed to a deep belief network to model fault detection and diagnosis. The experiments are performed on quantum computing devices. Progress on RBMs/GRBMs could potentially benefit such interdisciplinary research.

4. **Diversity of Generative Modelling Techniques**. The great progress of generative modeling research has benefited from a wide spectrum of techniques, including VAEs, GANs, EBMs, Normalizing Flows, Diffusion/Score-based models, and so on. Many of these models do not achieve state-of-the-art performances in their early stages, but still, make important contributions to the area by providing foundations for follow-up works. We believe our work on GRBMs would help increase the diversity of generative modeling techniques and stimulate more future work.

[1] Ajagekar, A. and You, F., 2020. Quantum computing assisted deep learning for fault detection and diagnosis in industrial process systems. Computers & Chemical Engineering, 143, p.107119.

[2] Melko, R.G., Carleo, G., Carrasquilla, J. and Cirac, J.I., 2019. Restricted Boltzmann machines in quantum physics. Nature Physics, 15(9), pp.887-892.

[3] Kieferová, M. and Wiebe, N., 2017. Tomography and generative training with quantum Boltzmann machines. Physical Review A, 96(6), p.062327.

---

### Decision · Program_Chairs · 2023-01-20

**Decision:**

Reject

**Justification For Why Not Higher Score:**

Weakness

---" I find the motivation on why using GRBMs for generative modeling and when they are preferable to other models is somewhat lacking. "
--- Below is AC's summary based on some discussions with the reviewer. One way to phrase the critical question is: how do we interpret the claim "the first time that GRBMs can generate images unconditionally"? Does GRBMs sound anywhere plausible for some future applications? We have to ask these questions because if one restricts to a sufficient small scope (in this case, only use GRBMs), then it's perhaps always possible to be the first. On the other hand, we also want to be open-minded enough to potential progress and realize that many breakthroughs didn't sound very promising at the early stage of the research. Unfortunately, I think the AC and the reviewers were generally not sufficiently convinced here.
The authors bring up some nice hypothetical possibilities enabled by their work but didn't have any solid applications. It appears to the AC and the reviewers that it would be good to have some solid (maybe new) applications that cannot be achieved with other EBMs or generative models.
--- there were other discussions on the (lack of) novelty of the work. However, it's not the major consideration for the decision.


**Justification For Why Not Lower Score:**

n/a

**Metareview: Summary, Strengths And Weaknesses:**

Strength:

---" The work improves on GRBM learning and introduces the first RBM method that can generate images from scratch. "
---" Sample quality is reasonable given the restrictions of the potential energy. "
--- "The theoretical insights in the paper were sound and important. "
--- "This is also the first time that GRBMs can generate images unconditionally. Although the sample qualities can be improved, the fact that this could be done is impressive even with a small model architecture is impressive."
--- some reviewers also acknowledge from their own experience that it's tough to make GRBM work.

Weakness

---" I find the motivation on why using GRBMs for generative modeling and when they are preferable to other models is somewhat lacking. "
--- Below is AC's summary based on some discussions with the reviewer. One way to phrase the critical question is: how do we interpret the claim "the first time that GRBMs can generate images unconditionally"? Does GRBMs sound anywhere plausible for some future applications? We have to ask these questions because if one restricts to a sufficient small scope (in this case, only use GRBMs), then it's perhaps always possible to be the first. On the other hand, we also want to be open-minded enough to potential progress and realize that many breakthroughs didn't sound very promising at the early stage of the research. Unfortunately, I think the AC and the reviewers were generally not sufficiently convinced here.
The authors bring up some nice hypothetical possibilities enabled by their work but didn't have any solid applications. It appears to the AC and the reviewers that it would be good to have some solid (maybe new) applications that cannot be achieved with other EBMs or generative models.
--- there were other discussions on the (lack of) novelty of the work. However, it's not the major consideration for the decision.